# Impact of drugs and environmental contaminants on amine production by gut bacteria

Stephan Kamrad (ID), Tara F Davis & Kiran R Patil (ID) ✉

## Abstract

**Xenobiotics like drugs are recognised as key influencers of gut bacterial growth. Yet, their impact on the production of metabolites involved in microbiota–host interactions is largely unknown. Here, we report the impact of commonly ingested xenobiotics—therapeutic drugs, pesticides, industrial chemicals, and sweeteners—on gut bacterial amine metabolism. We tested >13,000 interactions between >1700 compounds and 4 amine-producing bacteria, uncovering 747 xenobiotic-species-metabolite interactions involving 275 compounds. These compounds span all tested classes, with the majority being antimicrobial drugs. In 66% of the cases, amine production was correlated with growth, while the rest showed xenobiotic-induced decoupling between growth and metabolite production. The latter includes transient bursts in polyamine production by *Escherichia coli* in response to β-lactam antibiotics, and overproduction of aromatic amines by *Ruminococcus gnavus* treated with 15 diverse chemicals. Xenobiotics thus can disrupt metabolic homeostasis in both growth-dependent and -independent manner. We also find that metabolic responses have non-monotonic dose-dependency, resulting in lower doses sometimes having stronger effects. Our results bring forward the potential of common xenobiotics to disrupt the amine metabolism of gut bacteria.**

**Keywords** Microbiome; Xenobiotics; Pesticides; Stress Response; Polyamines
**Subject Categories** Digestive System; Metabolism; Microbiology, Virology & Host Pathogen Interaction

## Introduction

Small molecule metabolites produced by gut bacteria majorly contribute to human physiology and disease (Donia and Fischbach, 2015; Ma et al, 2022; Bishai and Palm, 2021). Among these, biogenic amines, produced by the decarboxylation of amino acids, are increasingly recognised as key effector molecules (Tofalo et al, 2019; Pugin et al, 2017; Sudo, 2019). The aromatic amines 2-phenylethylamine (PEA), tyramine and tryptamine (Sugiyama et al, 2022; Williams et al, 2014) act in the gut–brain axis and are

implicated in various diseases. Tryptamine and PEA produced by *Ruminococcus gnavus* have been mechanistically linked to insulin resistance in irritable bowel syndrome (IBS) and type-2 diabetes (T2D) patients (Zhai et al, 2023). Furthermore, PEA acts on the central nervous system (CNS) as a stimulant (Pei et al, 2016) and, in specific circumstances, as a toxin (Chen et al, 2019). In the gut, tryptamine has been shown to impact serotonin synthesis and gut motility (Bhattarai et al, 2018; Roager and Licht, 2018), as well as microbiome composition and metabolism (Otaru et al, 2024). Histamine, another biogenic amine, is an important signalling molecule during immune responses (Branco et al, 2018) and in the CNS (Haas et al, 2008). While human cells also produce it, bacteria-derived histamine has been shown to enhance gut motility (Chen et al, 2019), can act on distant parts of the immune system such as in the lung (Barcik et al, 2019), and is associated with irritable bowel syndrome (IBS) (De Palma et al, 2022; Mou et al, 2021). Finally, polyamines, containing two or more amine groups, are essential metabolites in humans and many other organisms, with great enzymatic and metabolic diversity among bacteria (Minguet et al, 2008; Kurihara, 2022), and a range of cellular and organismal functions (Miller-Fleming et al, 2015; Xuan et al, 2023). Polyamines in the gut have been shown to affect the immune system and the intestinal barrier (Nakamura et al, 2021), and might be positively associated with longevity (Matsumoto et al, 2011; Kibe et al, 2014).

For the producing bacteria, amines serve a range of beneficial functions. Generally, amino acid decarboxylation reactions consume protons (and release carbon dioxide) thereby raising the pH, which can be beneficial under acid stress conditions (Hersh et al, 1996; Richard and Foster, 2003; Gong et al, 2003; Schumacher et al, 2023b). Polyamines also protect cells against other stresses, such as oxidative stress (Solmi et al, 2023; Olin-Sandoval et al, 2019), osmotic stress (Schiller et al, 2000) and DNA damage (Iacomino et al, 2014; Terui et al, 2018; Gyu, 1998). Mechanistically, polyamines have high affinity for nucleic acids and, through interaction with ribosomes and tRNA, modulate translation (Winther et al, 2021; Dever and Ivanov, 2018). Furthermore, amine production and transport can generate proton motive force (Soksawatmaekhin et al, 2004), ultimately contributing to ATP production. Amines are also used by some bacteria as quorum sensing and biofilm molecules, with important implications for their survival and pathogenicity (Banerji et al, 2021; Nanduri and Swiatlo, 2021).

The importance of amines for both the microbiota and the host raises the question about the factors that could modulate their production. Gut bacteria are constantly exposed to an increasing

The Medical Research Council Toxicology Unit and Department of Biochemistry, University of Cambridge, Cambridge, UK. ✉E-mail: kp533@cam.ac.uk

diversity and level of xenobiotics, which are emerging as modulators of microbiome composition and function (Lindell et al, 2022). Pharmaceutical drugs are now recognised as one of the top factors influencing microbiome variation (Falony et al, 2016; Zhernakova et al, 2016; Forslund et al, 2021; Nagata et al, 2022). Less is known of the impact of agricultural and industrial chemicals which end up in food and drinking water. The extremely long half-life of some pollutants is resulting in their pervasive presence across the environment and food chain (Maggi et al, 2023; Egli et al, 2023) and the planetary boundary for human-made chemicals ('novel entities') is now believed to have been exceeded (Cousins et al, 2022; Steffen et al, 2015; Persson et al, 2022). Our recent data shows extensive impact on the growth of gut microbiota, suggesting that exposure to these compounds may also play a role in microbiota modulation (Lindell et al, 2024).

The unavoidable and pervasive exposure to diverse xenobiotics raises concerns around their impact on the metabolic function of gut bacteria. Anecdotally, xenobiotics have been shown to impact bacterial metabolism: the antidepressant drug duloxetine binds to cytoplasmic enzymes and significantly shifts the secretion of several metabolites impacting fellow community members (Klünemann et al, 2021), the anti-inflammatory drug sulfasalazine increases butyrate production in *Faecalibacterium prausnitzii* (Lima et al, 2024) and the antibiotic sulfamethoxazole stimulates the production of colipterins (Park et al, 2020). Yet, these examples span only a small fraction of the hundreds of thousands of xenobiotics that the microbiota is exposed to over lifetimes (Lindell et al, 2024).

Previous large-scale studies on xenobiotic-bacteria interactions have focused on growth effects (Maier et al, 2018; Lindell et al, 2024). Only a few studies have systematically investigated effects on bacterial physiology, notably one study investigating transcriptomic responses to >400 drugs (Ricaurte et al, 2024) and another investigating metabolic responses to 18 pesticides (Chen et al, 2025). We here investigated the effect of a large number of xenobiotics on gut bacterial amine metabolic output. Our results provide the first large-scale, pan-xenobiotic map of xenobiotic-species-metabolite interactions and highlight the potential of diverse xenobiotics to perturb amine production by gut bacteria.

# Results

## Bacterial species and target compounds

We selected four human gut bacteria based on documented ability to produce health-linked amine metabolites (Fig. 1A), viz., *Escherichia coli, Klebsiella aerogenes, Clostridium sporogenes* and *Ruminococcus gnavus. E. coli* is the archetypal polyamine producer in the gut (Kurihara, 2022). The strain used in this study (IAI1) was isolated from human faeces and produces cadaverine and putrescine when grown anaerobically in modified Gifu anaerobic medium (mGAM). The closely related pathogen *K. aerogenes*, from the same family (Enterobacteriaceae), is enriched in IBS patients (De Palma et al, 2022) and produces histamine, as well as polyamines. We also included two species from the phylum of Firmicutes, viz., *C. sporogenes* and *R. gnavus*. Both encode aromatic amino acids decarboxylases (AADC), and produce tryptamine in our culture conditions, with *R. gnavus* additionally producing 2-phenylethylamine (PEA).

## Compound library and methods overview

We employed throughput-optimised, targeted liquid chromatography—tandem mass spectrometry (LC-MS/MS) (Fig. 1B; Dataset EV1, 'Methods') to assess amine production in the presence of 1772 xenobiotic compounds (Dataset EV2), including 1518 pharmaceutical drugs, 166 agricultural pesticides, 47 industrial chemicals and pollutants, as well as 41 sweeteners (Fig. 1C). Bacterial cultures were grown in vitro in the presence of 20 μM of each compound, except for the low-calorie sweeteners where 50 μM was used as these are often consumed in larger quantities (Plaza-Diaz et al, 2020). This concentration represents a realistic gut exposure level for pharmaceutical drugs (Maier et al, 2018) and was chosen to facilitate comparisons to previous studies (Lindell et al, 2024; Maier et al, 2018; Garcia-Santamarina et al, 2024). After the stationary growth phase was reached, bacterial biomass was estimated using optical density measurement and metabolomics samples were prepared from the whole culture by removing proteins and inorganic salts with organic solvents. Two independent biological replicates for each compound were prepared on separate days.

## Quality indicators of metabolomics data

Measurements were divided up across a total of 24 batches. A total of 21,589 samples were acquired, including calibration standards, QC samples and blanks. Batch correction methods were used to correct for within- and across-batch variation ('Methods'), and quality control graphs were inspected (Fig. EV1A–D). Corrected metabolite concentrations exhibited low noise levels with a median coefficient of variation of 9.4% across DMSO controls (Fig. EV1E–H). Two independent biological replicates generally correlated well with a median Pearson correlation coefficient of 0.81 (Fig. EV1I–K). Overall, these quality indicators are in the typical range of biological mass spectrometry and indicate high data consistency despite very fast gradients and large batch sizes.

## Widespread impact of xenobiotics on amine metabolism

Robust and substantial production of amines was observed for all amine-bacteria pairs at baseline conditions (DMSO-treated control cultures), with concentrations ranging from 35 to 1100 μM (Fig. EV1F). This production capacity is significant in the context of the nanomolar-range physiological concentrations of these amines in the human body (D'Andrea et al, 2014; Rehn et al, 1987; Saito et al, 1983; Huebert et al, 1994) and the ability of aromatic trace amines to exert strong effects at very low concentrations (Burchett and Hicks, 2006; Zhai et al, 2023).

We defined significant xenobiotic-bacteria interactions where both biological replicates showed a consistent and statistically significant change ($P_{adj} < 0.05$, FDR-corrected z-test, 'Methods') and an absolute mean $\log_2$(fold change) >0.32 in metabolite concentration relative to DMSO controls (Dataset EV3). *R. gnavus* amine output is sensitive to the most xenobiotic compounds (196), followed by *C. sporogenes* (155), *E. coli* (132) and finally *K. aerogenes* (64) (Fig. 2A). Consistent with previous investigations of xenobiotic-bacteria interactions (Maier et al, 2018; Lindell et al, 2024), most compounds (1497/1772, 84%) have no effect on amine production by any species and broad spectrum activity is rare with

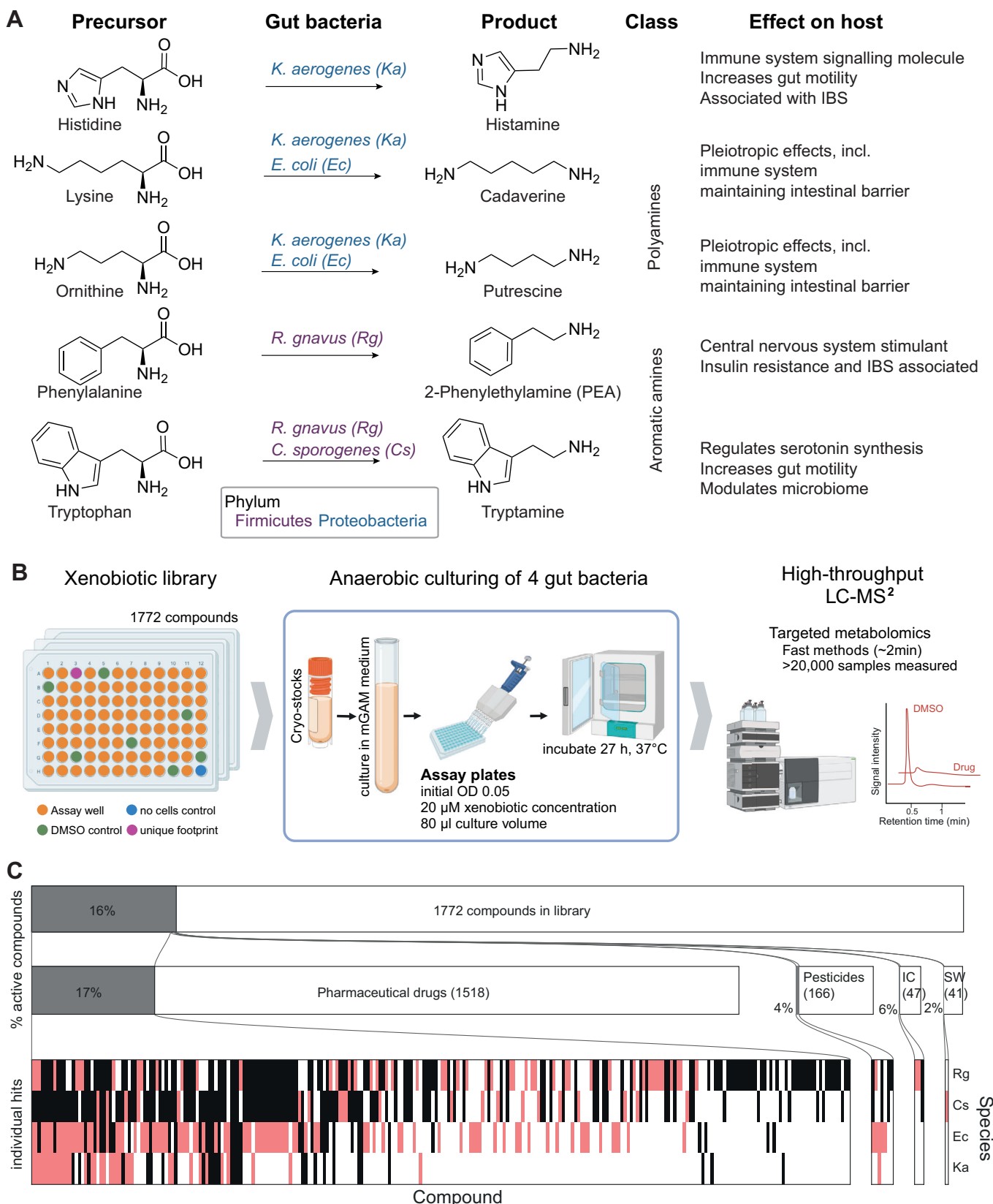

**Figure 1. Targeted metabolomics detects the impact of diverse xenobiotics on the amine metabolism of gut bacteria.**

(A) Overview of gut bacterial species used in this study, their target metabolites and documented effects on the host. Amino acid precursors are transformed into amines by bacterial decarboxylases. Decarboxylase repertoires -and amine production profiles- differ between species. Amines exert a wide range of effects on the host and the microbiome. Chemical structures were drawn with ChemDraw 23.1.2. (B) High-throughput workflow to assess the effect of xenobiotics on gut bacterial amine metabolism. Individual bacterial strains were cultured anaerobically in vitro in the presence of 20 μM of a library of diverse xenobiotics. At the stationary phase, growth was assessed by optical density measurement and metabolomics samples were extracted from the whole culture by the addition of organic solvent. Targeted metabolomics with fast gradients was used to determine amine concentrations in the presence of xenobiotics versus DMSO controls. Created in Biorender.com (License: Kamrad, S. (2025), BioRender.com/30el7cs). (C) Overview of results. Top: 16% of compounds across the entire chemical library affected the concentration of at least one amine in at least one bacterial species ($P_{adj} < 0.05$ (two-sided FDR-corrected z-test) and consistent direction of change in both biological replicates, abs ($\log_2$(fold change) >0.32, $n = 2$). The vast majority of hit compounds were pharmaceutical drugs (Prestwick library). Both the number of library compounds as well as the hit rate were lower for pesticides, industrial chemicals (IC) and low-calorie sweeteners (SW). The number of compounds in each compound class is indicated in parentheses, and the hit rate is indicated as a percentage. Bottom: The heatmap indicates significant individual compound-bacteria interactions and distinguishes between growth-concordant (absolute difference between mean growth fold change and mean metabolite fold-change <0.25) and non-concordant hits.

only 155 (8.7%) compounds affecting 2 or more species (Fig. 2B,C). Pharmaceutical drugs both have the highest hit rate (17%) and the highest overall number of hits, while other compound hit classes have hits of 6% or lower (Fig. 1C). Overall, these observations indicate a high degree of specificity in the susceptibility to xenobiotic perturbations.

We further explored this by classifying the hits based on direction of change, i.e., metabolite production increasing ('up hits') or decreasing ('down hits') (Fig. 2C). For *R. gnavus*, *C. sporogenes* and *K. aerogenes*, down hits are far more common, but for *E. coli*, they are approximately balanced with 75 and 92 hits, respectively. Next, we investigated the relationship between growth and amine production by assessing if the change is concordant with the change in growth (absolute difference between mean growth fold change and mean metabolite fold change <0.25). Growth inhibition results in lower bacterial biomass with an expected concurrent reduction in metabolic activity. However, we observed a wide range in the degree of coupling between relative growth and relative amine concentration (both relative to plate-matched DMSO controls) (Fig. 2D), ranging from 86% of hits behaving concordantly for tryptamine production by *C. sporogenes* (Fig. 2E) to 12% for cadaverine production by *K. aerogenes*. *E. coli* similarly exhibited low growth concordance (Fig. 2F) and the overall highest number of non-concordant hits. This indicates a species-dependent decoupling of amine metabolism and growth under xenobiotic stress, with particularly polyamine biosynthesis in *E. coli* and *K. aerogenes* showing a large degree of decoupling.

## Antibacterial drugs decouple amine production and growth

As pharmaceutical drugs were responsible for the vast majority of hits (Fig. 1C), we next tested for specific groups of drugs that are statistically overrepresented. Antibiotics (antibacterials) are widely enriched across species as well as in growth-concordant and non-concordant metabolic changes (Fig. 2G) (FDR-corrected Fisher's Exact test, $P_{adj} < 0.05$), including those where increased metabolite production without a concordant increase in growth was observed in *E. coli*. The hits in the firmicutes *R. gnavus* and *C. sporogenes* are also enriched in antiprotozoals, antiamoebics and antifungals, with these causing a decrease in growth and a concordant decrease in amine production. Complementarily, we also computed hit rates (number of compounds in class affecting at least one bacterial metabolite/number of compounds in class) for each therapeutic effect class and similarly found antimicrobials such as

antibacterials, antiprotozoans, antiseptics and antifungals to have the strongest effects (Fig. 2G).

## Divergence of metabolic responses in Enterobacteriaceae

Since *E. coli* and *K. aerogenes* are closely related and members of the same family, we expected similar metabolic responses to xenobiotic exposure. This is consistent with previous results showing a phylogenetic contribution to xenobiotic resistance profiles (Lindell et al, 2024; Maier et al, 2018; Ricaurte et al, 2024). Indeed, while a strong correlation (Pearson $r = 0.79$) was observed at the level of growth (Fig. 3A), a much weaker correlation was observed for the two polyamines produced by both species, cadaverine ($r = 0.35$, Fig. 3B) and putrescine ($r = 0.43$, Fig. 3C). This indicates that metabolic responses differ even between closely related species and evolve faster than susceptibility phenotypes at the growth level.

## Non-monotonic dose-dependency of xenobiotic–amine metabolism interaction

To assess how the metabolic response changes with compound concentration, we tested metabolic and growth responses at a range of concentrations between 0.6 and 20 μM (Dataset EV4). For this, we chose a subset of 10–33 xenobiotics that showed heterogeneity in the growth response per species. The fraction of compounds stimulating amine metabolism changes non-linearly along the concentration gradient (Fig. 3D). For putrescine production by *E. coli*, 5 of the 13 (38%) compounds tested elicit an increased production compared to DMSO control when applied at 2.5 μM, versus 2 compounds (15%) applied at 20 μM. For tryptamine production by *R. gnavus*, the strongest stimulatory effect was observed at 2.5 and 5 μM. At the level of individual compounds, we observed typical, monotonous dose–response behaviour for growth (i.e., higher treatment concentration equals stronger response) but atypical behaviours at the metabolic level (medium doses usually resulted in the strong response) (see Fig. 3E,F for two examples, and Fig. EV2 for details).

## β-lactams stimulate polyamine production in *E. coli*

*E. coli* showed by far the most instances of increased metabolite production, and the vast majority of these (67/85) were cadaverine. Out of 60 xenobiotics which stimulated cadaverine production by *E. coli*, 55 were pharmaceutical drugs, and out of those, 38 (69%)

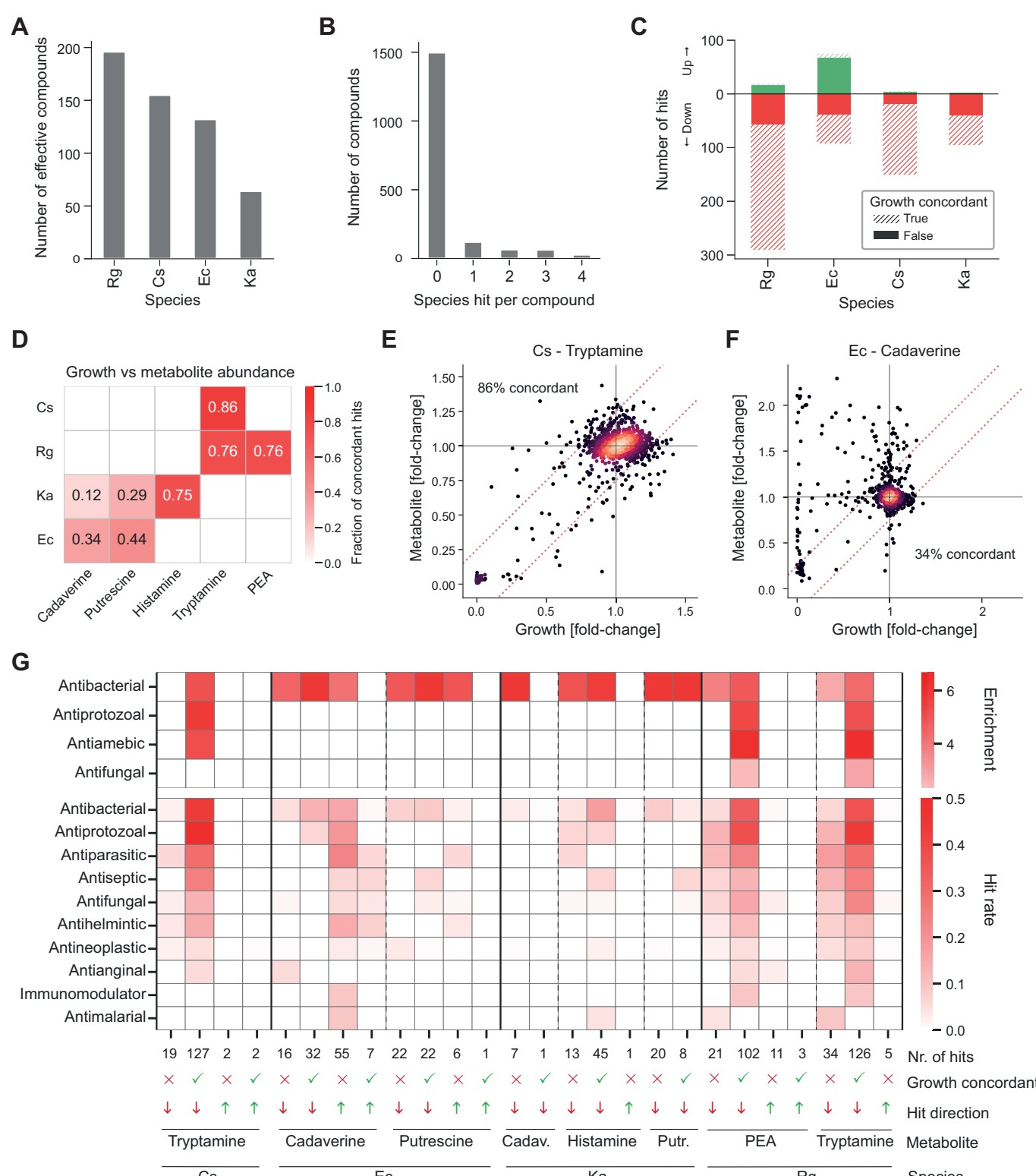

were annotated as antibacterials (Fig. 4A). Further dissecting the effect of different antibiotics classes (annotation from (Maier et al, 2021)), we identified 24 β-lactam antibiotics, which inhibit biosynthesis of the cell wall, as the main stimulators of cadaverine metabolism (Fig. 4B). Fluoroquinolones, disrupting DNA, and

tetracyclines, disrupting translation, mostly showed strong, growth-concordant reductions in cadaverine production. Other antibiotic classes like macrolides and aminoglycosides (also both translation inhibitors), and sulphonamides (targeting folate metabolism) showed little effect on both metabolism and growth, consistent

**Figure 2. Xenobiotic exposure can uncouple amine metabolism and growth.**

(A) Number of compounds which affect at least one amine metabolite, by species. (B) Distribution of the number of species against which each compound is effective. While 84% of the compounds do not affect the amine output of any species, 8.7% affect two species or more. (C) Number of hits with increased or decreased amine metabolite concentrations, further distinguishing hits whether hits were concordant with growth (absolute difference between mean growth fold-change and mean metabolite fold change below 0.25). (D) Heatmap showing the fraction of growth concordant changes across different bacteria and amine metabolites. PEA 2-phenylethylamine. (E) Tryptamine production by *C. sporogenes* is tightly coupled to growth. (F) Cadaverine production by *E. coli* is largely uncoupled from growth. In many instances, cadaverine production is substantially increased while growth is either unaffected or reduced. (G) Heatmaps illustrating the activity of drugs with different therapeutic effects, as annotated in the Prestwick library. Top: Enrichment of drug classes across different species, metabolites, hit direction and growth concordance ($x$ axis). Only significantly enriched terms ($P_{adj} < 0.05$, Fisher's exact test) and positive enrichments (hit frequency > background frequency) are shown. Bottom: Hit rate (number of active compounds / number of compounds in class) for top ten drug classes.

with natural and acquired resistance of *E. coli* to these compounds (Ma et al, 2024; Ojdana et al, 2018; Venkatesan et al, 2023).

While it is not unexpected that antibiotics trigger polyamine production (a common stress response mechanism), we were intrigued by the differences between antibiotic classes and the general lack of published data investigating the physiological impact of antibiotics in anaerobic environments. We therefore independently validated and further investigated the dose-dependency of a selected number of antibiotics from different classes on cadaverine production (Fig. EV3A). The three β-lactams tested indeed stimulated cadaverine production and this was again strongest in the low micromolar range. Representatives from two other antibiotic classes, tetracycline and levofloxacin (a fluoroquinolone) did not trigger cadaverine production at any dose.

Antibiotics are often distinguished into bacteriostatic and bactericidal antibiotics depending on whether they kill bacterial cells or just arrest their growth, although this classification is not strict across species and doses (Maier et al, 2021). For the five antibiotics shown in Fig. EV3A, we found no link between static or cidal activity and cadaverine production, as all β-lactams were cidal against our *E. coli* strain and only tetracycline treatment left some viable cells (Fig. EV3B). We then assessed the temporal dynamics of growth and cadaverine production in the presence of the β-lactam ceftazidime (Fig. 4C). Treated cultures showed a brief period of growth with a concurrent burst in cadaverine production, indicating that there is a brief period of time where cells can grow and produce large amounts of cadaverine before being killed by the antibiotic.

To further explore the role of polyamine metabolism under antibiotic treatment, we re-analysed a publicly available dataset (Noto Guillen et al, 2024) that captures the fitness of pooled, genome-wide knock-out mutants in the presence of numerous antibiotics and non-antibiotics. Focusing on the 8 key polyamine-producing enzymes (Chattopadhyay et al, 2009), the data indicates numerous abundance changes of the corresponding knock-out strains in pooled cultures compared to untreated controls, however no clear pattern that distinguishes antibiotic classes or polyamine pathways is apparent. Surprisingly, mutants of *speA* (arginine decarboxylase) and *speB* (agmatinase), enzymes producing agmatine and putrescine, often have significantly improved fitness in the mutant pool when challenged with diverse antibiotics, suggesting that these functions are not required for improved growth/survival or compensated by polyamine sharing within the pooled culture (Fig. EV3C).

### Non-antibacterials stimulate amine metabolism

While the set of hit compounds which increase polyamine production in *E. coli* is dominated by drugs with known antibacterial effects, several non-antimicrobial drugs show similar effects, indicating that amine-stimulation is not unique to antibiotics. Substantial increases in cadaverine production were observed and independently validated with the pesticide fenazaflor (Fig. 4D), the HIV drug didanosine (Fig. 4E) and the selective serotonin reuptake inhibitor antidepressant paroxetine (Fig. 4F). Only fenazaflor causes a substantial (~40%) change in growth. This indicates that non-antibiotic drugs have the potential to induce polyamine metabolism in *E. coli*, similarly to well-known stressors such as antibiotics.

To further explore the mechanism of increased cadaverine production, we mined a published dataset of transcriptional responses by diverse gut bacteria to selected pharmaceutical drugs (Ricaurte et al, 2024). Under paroxetine treatment, the cadaverine biosynthesis genes *cadA* and *cadB* and the acid stress response gene *asr* were strongly upregulated (Fig. 4G,H). The putrescine:proton symporter *plaP* and a lysine permease *lysP* were also upregulated, albeit weaker (Fig. 4H). This is in alignment with the observed metabolic phenotype and provides a mechanistic link via a transcriptional stress response.

### Mono-amines are stress-induced metabolites in *R. gnavus*

After *E. coli*, *R. gnavus* showed most up hits, with a total of 19 significant xenobiotic-metabolite interactions, originating from 15 unique compounds. All except one (1,4-Benzoquinone, an industrial chemical) were pharmaceutical drugs. Among the 8 compounds triggering amine increases of 50% or more were etanidazole (a disused investigational oncology drug), liranaftate (a topical antifungal), enilconazole (an agricultural and veterinary antifungal), rebamipide (used to treat gastritis), sulfadimethoxine (a veterinary antimicrobial) and balsalazide (used to treat inflammatory bowel disease (IBD)). These results indicate that mono-amines such as PEA and tryptamine can fulfil similar physiological roles as polyamines and are produced in response to xenobiotic stress in *R. gnavus* but not the related species *C. sporogenes*.

## Discussion

Our study provides the first pan-xenobiotic scale overview of the relationship between growth and metabolic output in anaerobically cultured gut bacteria, providing systematic evidence that xenobiotics have effects beyond the growth of bacteria. We find that many compounds, particularly antibacterials, can uncouple amine production from growth and cause an increase in polyamine production in *E. coli*.

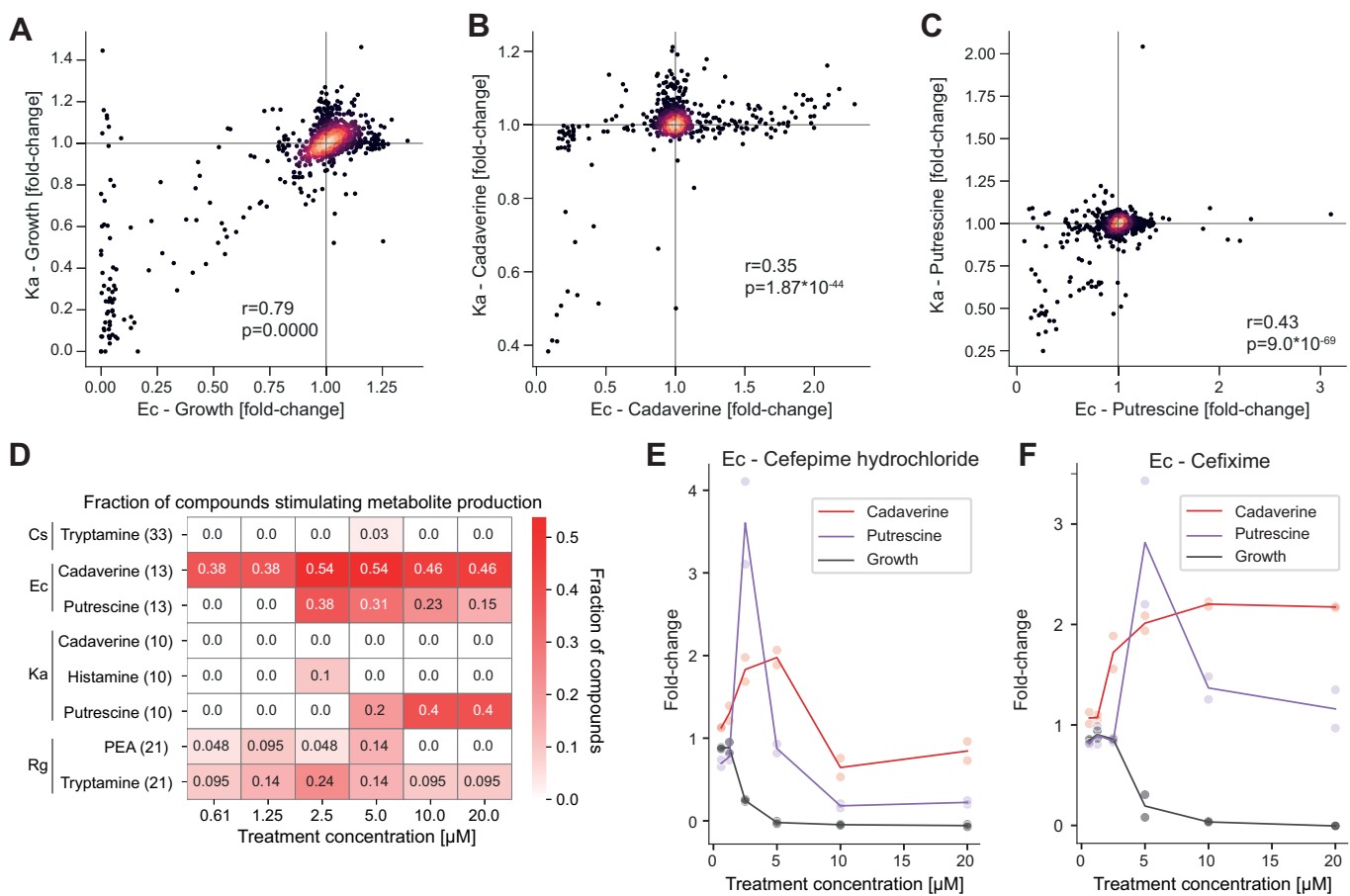

**Figure 3.  Non-monotonous dose-dependency of growth and amine metabolism.**

(A) The growth response to xenobiotics correlates well between the two closely related Enterobacteriaceae *E. coli* and *K. aerogenes*. The Pearson correlation coefficient and associated two-sided *P* value is indicated in the plot for (A–C). (B) The cadaverine production response is correlated to a much weaker degree. (C) Same as above, for putrescine. (D) A subset of pharmaceutical drugs was tested at a concentration range between 0.6 µM and 20 µM (the concentration of the main screen). Amine production was assessed for compounds which showed a varied growth response (i.e., no growth inhibition at low concentrations, strong growth inhibition at high concentrations. The number of compounds tested is indicated behind the species name (*y* axis). The fraction of compounds which stimulated a statistically significant increase in amine production ($P_{adj} < 0.05$ (two-sided FDR-corrected z-test) for both biological replicates). Mild xenobiotic stress at 2.5–5 µM showed the strongest stimulation of amine metabolism in *E. coli* and *R. gnavus* (tryptamine only). This indicates that xenobiotics can stimulate amine metabolism at a range of doses and that this does not follow a typical dose–response expected at the level of growth (higher concentration equals stronger effect). (E, F) Examples of dose–response curves in *E. coli* for two commonly used β-lactam antibiotics, illustrating the unusual dose–response behaviour. The lines indicate the mean and the shaded areas indicate the standard deviation of $n = 2$ biological replicates.

Previous work has established polyamines as stress response metabolites in the context of acid and osmotic stress (Gong et al, 2003; Tofalo et al, 2019; Noack et al, 1998). Others have shown that some antibiotics can affect metabolism in *E. coli* (Zampieri et al, 2017) and induce polyamine production (Tkachenko et al, 2012), and that cadaverine production contributes to antibiotic resistance (Akhova et al, 2021). These studies used aerobic cultures and attributed this effect to the formation of reactive oxygen species (ROS). Other studies (also in aerobe conditions) have linked antibiotics to acid stress (Schumacher et al, 2023a): In *Pseudomonas aeruginosa*, the aminoglycoside amikacin reduces cytoplasmic pH (Arce-Rodríguez et al, 2019), and in *E. coli* trimethoprim induces the *gadBC* acid stress response operon (Mitosch et al, 2017). However, in *Mycobacterium smegmatis*, bactericidal antibiotics were found to raise intracellular pH (Bartek et al, 2016), indicating differences between bacterial species and/or

antibiotic classes. Our results in *E. coli* capture diverse antibiotics and non-antibiotic xenobiotics and show that polyamines are stress-responsive in the context of anaerobic metabolism (in the absence of ROS).

What is the mechanism causing polyamine upregulation in response to antibiotic exposure under anaerobic conditions? There remain significant knowledge gaps around the downstream mechanisms of action of bactericidal antibiotics in anaerobic conditions, specifically how inhibiting the direct molecular target (penicillin-binding proteins in the case of β-lactams) eventually leads to cell death. Reactive electrophilic species have been described to play a role in this context, with downstream effects including DNA and membrane damage (Wong et al, 2022), which in turn could induce polyamines.

While polyamines are known stress-induced metabolites, the role of mono-amines in stress response is less well described; the

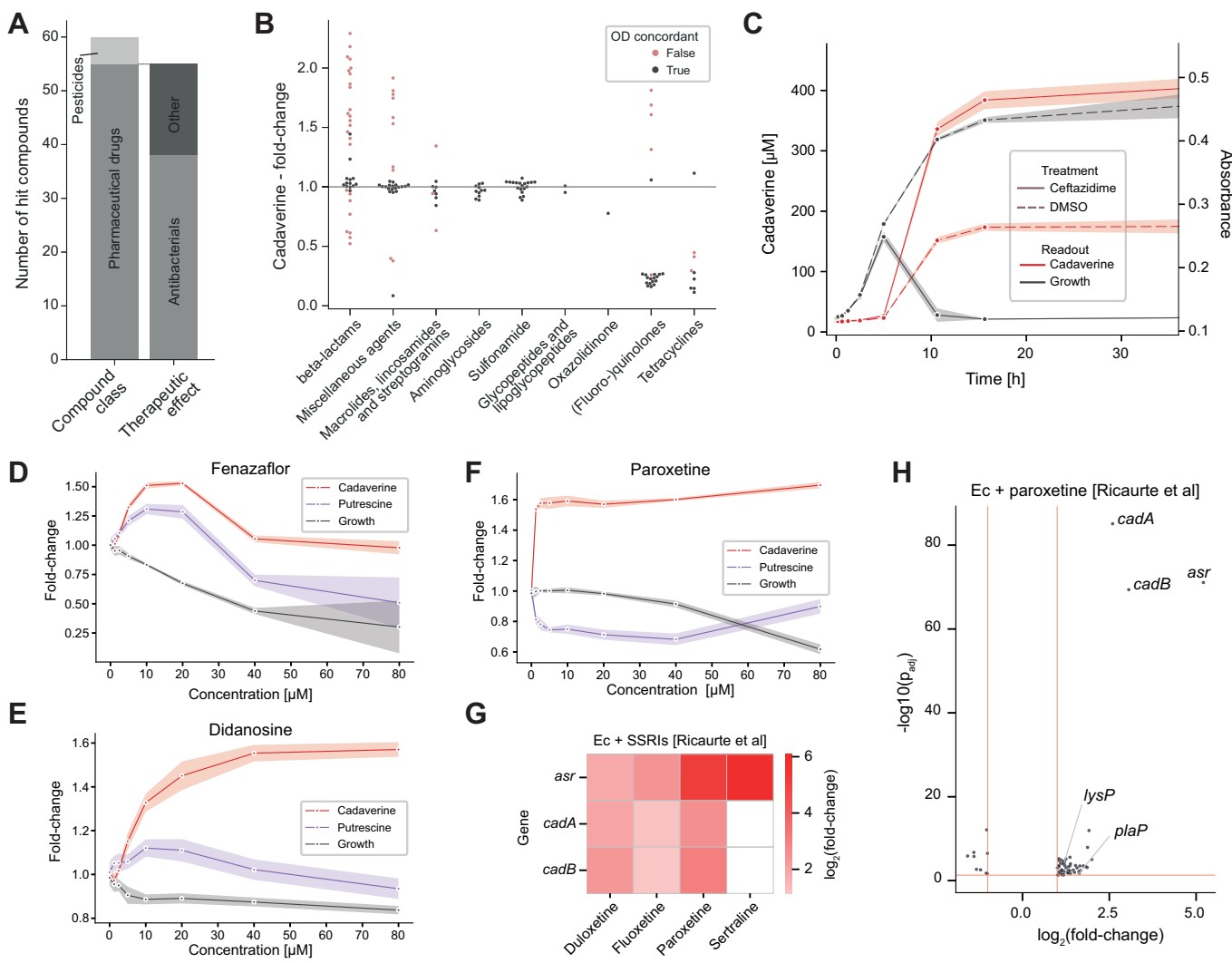

**Figure 4. β-lactam antibiotics trigger polyamine production by *E. coli*.**

(A) Focusing on xenobiotics which stimulate cadaverine production in *E. coli*, the vast majority of these were pharmaceutical drugs with annotated antibacterial effect. (B) Effect of different antibiotic classes on cadaverine production by *E. coli*. β-lactams show the strongest effect with many stimulating cadaverine production in a manner not concordant with growth (absolute difference between mean growth fold change and mean metabolite fold change below 0.25). Each datapoint is the mean of $n = 2$ biological replicates. Annotation from (Maier et al, 2021). (C) Growth and cadaverine production dynamics in *E. coli* treated with 20 μM of ceftazidime. Treated cultures show a brief period where the optical density (Absorbance, 595 nm) increases, concurrent with a burst of cadaverine production far above untreated levels. This indicates that this β-lactam antibiotic does not immediately kill the cells but allows for a brief period of growth and stress-induced polyamine production prior to bacterial cell lysis. Lines indicate the mean, and shaded areas the standard deviation of $n = 6$ biological replicates. (D–F) Dose–response curves for fenazaflor (pesticide), didanosine (antiviral drug) and paroxetine (antidepressant), indicating fold changes of either metabolite concentrations (red and purple) or growth at the stationary phase, compared to untreated controls. Lines indicate the mean and shaded areas indicate the standard deviation of $n = 8$ biological replicates. (G) Re-analysis of data from (Ricaurte et al, 2024) illustrating transcriptomic regulation of polyamine biosynthesis genes in *E. coli* MG1655 in the presence of different selective serotonin reuptake inhibitor (SSRI) antidepressant drugs. Data was obtained from Supplementary Table 8. (H) Volcano plot illustrating genome-wide transcriptional changes in *E. coli* MG1655 in the presence paroxetine, obtained from Supplementary Table 8 of (Ricaurte et al, 2024). Differentially abundant transcripts with known roles in polyamine metabolism are annotated in the plot. Red lines indicate the fold-change thresholds used in the original publication ($log_2$(fold change) >1 and $P_{adj} < 0.05$ (FDR-corrected $P$ values obtained via DESeq2)).

decarboxylation of glutamate to form γ-aminobutyric acid under acid stress in *E. coli* is a notable exception (Hersh et al, 1996). In particular, aromatic amino acid-derived amines are not known stress-induced metabolites. For example, tryptamine was not found to be induced by acid stress (Otaru et al, 2024). We here show that 'trace' amines, in particular tryptamine and PEA, which are active on the gut and central nervous system at very low concentrations (Sherwani and Khan, 2016), are xenobiotic-responsive metabolites.

They might therefore fulfil similar biological roles in species which possess aromatic amino acid decarboxylases but do not produce polyamines.

Our data also reveal striking differences between the closely related Enterobacteriaceae *E. coli* and *K. aerogenes*, with the latter showing no evidence of polyamine overproduction upon xenobiotic stress, despite possessing the required pathways and producing basal levels of polyamines. This underlines the difficulty of

extrapolating biological findings using phylogenetic similarity, even within closely related species. In addition, in contrast to growth responses, metabolic responses follow unconventional dose–response patterns, with lower doses often eliciting stronger responses. In the case of β-lactam antibiotics and *E. coli*, this is due to the ability of more mildly stressed cells to survive xenobiotic stress for a small number of generations during which amine metabolism is strongly upregulated. Both these points should therefore be considered in toxicological assessment and predictions of the effect of xenobiotics on gut bacterial metabolic output.

Future studies will be needed to show if amines represent a 'special' class of metabolites in this respect or if other metabolite classes are similarly responding to specific xenobiotic stressors. Classically, a distinction is made between fermentation products (tightly coupled to growth) and other more peripheral metabolites (produced for other, secondary purposes such as stress response, defence or signalling) (Drew and Demain, 1977). However, this distinction is challenged by findings like the anti-inflammatory drug sulfasalazine increasing the production of the short-chain fatty acid butyrate (a 'primary' metabolite and fermentation product) (Lima et al, 2024).

In conclusion, our study represents a first large-scale map of xenobiotic–bacteria–metabolite interactions. Understanding the impact of xenobiotics on metabolism of gut bacteria could be key in several areas, including understanding the impact of unintentionally consumed pollutants/contaminants on microbiome metabolism, and the contribution of such metabolic effects to the mechanism of action or adverse effects of pharmaceutical drugs. Ultimately, targeted interventions could be used to shape metabolic outputs of the microbiome.

# Methods

### Reagent and tools table

| Reagent/resource | Reference or source | Identifier or catalogue number |
| --- | --- | --- |
| **Experimental models** | | |
| *Clostridium sporogenes* | DSMZ | DSM 1664 |
| *Escherichia coli* | Denamur Lab, INSERM | IAI1 |
| *Klebsiella aerogenes* | DSMZ | DSM 30053 |
| *Ruminococcus gnavus* | DSMZ | DSM 108212 |
| **Chemicals, enzymes and other reagents** | | |
| Modified Gifu anaerobic medium (mGAM) | Nissui Pharmaceuticals (sold by HyServe, Germany) | 1005433-001 |
| Clear untreated sterile 96-well polystyrene plates, u-bottom | Corning | 3795 |
| Pharmaceutical drug library | Prestwick Chemical Libraries | Dataset EV2 |
| Pesticide library | EMBL Chemical Biology core facility | Dataset EV2 |
| Industrial chemicals library | Lindell et al, 2024 | Dataset EV2 |
| Sweeteners library | Blasche et al, 2025 | Dataset EV2 |
| Ampicillin | Sigma-Aldrich/Merck | A9518 |

| Reagent/resource | Reference or source | Identifier or catalogue number |
| --- | --- | --- |
| Cefixime | Cayman Chemical Company | 17176 |
| Ceftazidime | MedChemExpress | HY-B0593/CS-2810 |
| Levofloxacin | Cayman Chemical Company | 20382 |
| Tetracycline | Sigma-Aldrich/Merck | 87128 |
| Paroxetine | TargetMOl | T1636L |
| Didanosine | Cayman Chemical Company | 23715 |
| Fenazaflor | Sigma-Aldrich/Merck | 36504 |
| Acetonitrile | VWR | 83640.320 |
| Methanol | VWR | 83638.320 |
| Formic acid | Fisher | A117-50 |
| Ammonium formate | VWR | 84884.180 |
| Caffeine | Fluka | 56396 |
| Amoxicillin | Sigma-Aldrich/Merck | A8523 |
| Putrescine | MetaSci | Complete Metabolite Library |
| Cadaverine | Sigma-Aldrich/Merck | C8561-1G |
| Histamine | MetaSci | Complete Metabolite Library |
| Tryptamine | MetaSci | Complete Metabolite Library |
| 2-Phenylethylamine | MetaSci | Complete Metabolite Library |
| **Software** | | |
| MassHunter Workstation | Agilent | v10.1 |
| SkanIt Microplate Reader Software | ThermoFischer | v 6.1.1 |
| Python | | v 3.11.5 |
| pandas package | McKinney, 2010 | v. 2.2.0 |
| scipy package | Virtanen et al, 2020 | v. 1.11.3 |
| numpy package | Harris et al, 2020 | v. 1.26.0 |
| scikit-learn package | Pedregosa et al, 2012 | v. 1.3.2 |
| matplotlib package | Hunter, 2007 | v. 3.8.0 |
| seaborn package | Waskom, 2021 | v. 0.13.1 |
| **Other** | | |
| Analytical column: ACQUITY BEH Amide 1.7 μm, 2.1 × 50 mm | Waters | 186009504 |
| Analytical column: InfinityLab Poroshell 120 EC-C18 1.9 μm, 2.1 × 50 mm | Agilent | 699675-902 |
| Triple-quadrupole mass spectrometer | Agilent | 6470 |
| Ultra-high performance liquid chromatography instrument | Agilent | 1290 Infinity II |

| Reagent/resource | Reference or source | Identifier or catalogue number |
|---|---|---|
| Biomek Automated Workstation | Beckman Coulter | i7 |
| Microplate reader | ThermoFisher | Multiskan FC, 51119100 |

## Xenobiotic libraries

Xenobiotic compounds were obtained from various sources. The pharmaceutical drug library was obtained from Prestwick Chemical Libraries and contained 1518 approved drugs. A library of pesticides was assembled by the Chemical Biology Facility of EMBL Heidelberg and contained 166 commonly used pesticides. Libraries of industrial contaminants (47 compounds) and sweeteners (41 compounds) were prepared in house. Compound metadata including PubChem compound IDs and order numbers is available in Dataset EV2. All compounds were dissolved in DMSO and original stocks were concentrated at 10 mM. From these stocks, 2 mM working stocks (or lower for the dose–response screen) were prepared by dilution into DMSO and a tetracycline solution was added to a unique empty position in each plate. Any remaining empty positions were filled with DMSO and served as controls. Plates contained between 10 and 13 DMSO controls. Working stocks were then divided into 10 μL aliquots and frozen at −80 °C. Each day of the experiment, a compound aliquot plate was defrosted and its entire content diluted in 490 μL of media in a 96 deep-well plate, from which 96-well assay plates were filled with 40 μL.

## Bacterial cultivations

All bacteria used in this study are well characterised strains (usually type strains), mostly obtained from culture collections. All cultures were grown statically at 37 °C in an anaerobic polyvinyl chamber (Coy Instruments) filled with 2.5% $H_2$ and 12% $CO_2$ in $N_2$. Cultures were grown in modified Gifu anaerobic broth (mGAM) prepared according to the instructions from the manufacturer and sterilised by autoclaving. All media was introduced into the chamber at least 16 h before cells were added for it to become anaerobic.

1. From glycerol-preserved cryostocks, 10 mL cultures were grown in screw-top tubes for 1 or 2 days (depending on the growth rate of the bacterial species, but always consistent within the same species).
2. Cultures were diluted 100-fold into 10 mL of fresh media and incubated again for the same time.
3. The optical density (OD) was measured, cultures were diluted to OD 0.1 and 40 μL were transferred to the pre-filled 96-well plates containing media with xenobiotic compounds. The final culture volume was 80 μL per well and the final compound concentration was 20 μM.
4. Plates were sealed with aluminium seals and incubated for 27 h at 37 °C.

## Growth data analysis

After incubation, plates were removed from the anaerobic chamber and shaken for 10 seconds at 1000 rpm (Thermomix, Eppendorf).

Seals were removed and the OD was determined by measuring the absorbance at 595 nm in a microplate reader. Raw values were blank-subtracted. We also corrected for the background signal of xenobiotics (some of these produced coloured solutions which absorb at 595 nm) by subtracting blank-corrected absorbance values of 20 μM solutions prepared independently in water. Absorbance values were converted to conventional OD measurements by multiplying with 5 (based on a previously determined calibration curve for the same instrument). OD values were clipped at 0 to remove a small number of slightly negative values occurring when a xenobiotic strongly inhibited growth and the resulting absorbance fell slightly below blank levels due to measurement noise. Relative OD fold changes were obtained by dividing values by the median of the DMSO controls on the same 96-well plate. We observed a small number of wells (353 out of 4608, 7.7%) with unusually high OD fold-changes of 1.5 or higher. Over 80% of these belonged to *R. gnavus* and this was largely consistent across replicates of the same compound and is therefore not due to random effects (Fig. EV4A). Instead, we found that this is due to aggregation of cells which confounds OD measurements (Fig. EV4B,C), consistent with such observations in other species (Haaber et al, 2012). Growth fold changes above 1.5 were set to NA and excluded from further analysis. No blinding was done in this study.

## Metabolomics sample processing

After OD measurement, metabolomics samples were extracted from the same plates:

1. On a Biomek i7 Automated Workstation (Beckman Coulter), 120 μL of cold extraction solution (1:1 acetonitrile and methanol, with 0.1% formic acid, 15 μM caffeine and 10 μM amoxicillin) was added to each well.
2. Plates were sealed with an aluminium seal, shaken for 10 s as before and incubated at 4 °C for ~1 h.
3. Extracts were then cleared of cell debris and insoluble components by centrifugation (5 min, 2400 × g, 4 °C).
4. 20 μL of supernatant were aliquoted into 384-well PCR plates, stored at −80 °C until LC-MS analysis.

## Metabolomics measurements

Samples were measured on a liquid chromatography setup (Agilent 1290 Infinity II) coupled to a triple-quadrupole mass spectrometer (Agilent 6470 with JetStream electrospray ionisation source) operated in dynamic multiple reaction monitoring (dMRM) mode. A tailored acquisition method, mixed analytical standard of pure compounds and reference QC sample was prepared for each species (see Dataset EV1 for all parameters). We used reverse phase or hydrophilic interaction liquid chromatography (HILIC) depending on target compound properties. Multiple transitions were measured for most analytes to improve confidence in identification. Methods were optimised for fast runtimes to increase throughput. 0.5–1 μL of sample were injected and blocks of 12 samples were interleaved with blanks, a 3× dilution series with 6 levels of a mixed external standard and reference QC samples. Each measurement batch, consisting of 2–4 384-well plates, was analysed separately using Agilent MassHunter Quantitative Analysis

for QQQ (v. 10.1). Calibration curves were fitted based on external standards and used to convert peak areas to concentrations.

## Metabolomics data analysis

Data were processed in Python with standard packages (see 'Tools and Reagents table'). Within each batch, concentrations were normalised across samples to account for signal drift over time (Fig. EV1). For this, a trend line was fitted to the sample data (excluding standards, blank and QC samples) by first removing outliers (defined as in a standard boxplot, with bounds: 1st quartile—1.5× interquartile range; 3rd quartile + 1.5× interquartile range), and then fitting a local regression using the RadiusNeighborsRegressor function with radius = 20.0 from the scikit-learn package (radius = 30 for dose–response screen). For each datapoint, the deviation (ratio of observed value to trend line) from the fitted line was determined and a corrected concentration was estimated by multiplying with the median compound concentration across the entire screen. Corrected concentrations were clipped at 0.01 μM to remove negative and zero values. A z-score statistic was computed by dividing the difference between the observed corrected concentration from the 96-well plate median by the standard deviation of the corrected concentration of the DMSO controls on the same 96-well plate. The survival function of the normal distribution was used to convert z-scores to a $P$ value. Within each species and measured metabolite, $P$ values were corrected for multiple testing using the Benjamini–Hochberg method. A compound was considered a hit if $P_{adj} < 0.05$ in both biological replicates, both replicates showed the same direction of change and if the absolute $\log_2$-transformed mean fold change relative to 96-well plate median was greater than 0.32 (i.e., fold change outside the approximate range of 0.8–1.25).

## Data availability

The datasets and computer code produced in this study are available in the following databases: Targeted metabolomics data and associated scripts: Mendeley Data https://doi.org/10.17632/cds7tvdb85.2.

The source data of this paper are collected in the following database record: biostudies:S-SCDT-10_1038-S44320-025-00130-4.

## Peer review information

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

## Acknowledgements

The authors would like to thank Luisa Faria for help with 16S Sanger sequencing. This project has received funding from the European Research Council (ERC) under the European Union's Horizon 2020 research and innovation programme (grant no. 866028) (KRP and SK) and from the UK Medical Research Council (project no. MC_UU_00025/11) (KRP and TFD).

## Author contributions

**Stephan Kamrad**: Conceptualisation; Formal analysis; Supervision; Investigation; Visualisation; Methodology; Writing—original draft; Writing—review and editing. **Tara F Davis**: Investigation. **Kiran R Patil**: Conceptualisation; Supervision; Funding acquisition; Project administration; Writing—review and editing.

Source data underlying figure panels in this paper may have individual authorship assigned. Where available, figure panel/source data authorship is listed in the following database record: biostudies:S-SCDT-10_1038-S44320-025-00130-4.

## Disclosure and competing interests statement

The authors declare no competing interests.

# Expanded View Figures

**Figure EV1.  Quality control indicators.**

(A) In each LC-MS run, performance was monitored by regular injection of serially diluted analytical standards, a QC sample (DMSO-control sample from respective bacterial species) and water blanks. All runs were inspected manually and a technical coefficient of variation (CV, indicating the stability of instrument performance) was computed based on the distribution of values obtained for the QC sample. (B) Batch variation between each 384-well sample plate was apparent. This was corrected using the strategy described in Methods. In brief, a smoothed line (black) was fitted to outlier-filtered data and the distance of each datapoint to the line was added to the overall median to produce batch-corrected concentration estimates. (C) Concentration values after normalisation. (D) Hit calling was performed on each sample (ie biological replicate) separately. The deviation of the sample from the fitted line (see B) was divided by the standard deviation of the DMSO controls, for each 96-well culture plate (indicating the degree of combined biological and technical noise in the assay) separately. This z-score was converted to a p value using the normal distribution. (E) Distribution of fold changes (relative to mean of DMSO controls) for DMSO controls and xenobiotic-treated samples. As the vast majority of compounds had no effect on a bacterial species, the distributions are similar, although increased density around 0 is observed in treated samples. (F) Based on DMSO controls, the baseline median concentration (top) and the overall level of biological and technical noise in the assay (bottom) were quantified. CVs fell between 8 and 18%, a typical range observed in biological mass spectrometry. $N > 600$ biological replicates, error bars indicate the standard deviation. (G) Quantile-quantile (Q-Q) plot of fold-change values of DMSO controls across all species and amines. This indicates approximately normal distribution of the values, albeit with slightly heavy tails. (H) Power calculation of the Z-test used to determine likelihood that individual datapoints are compatible with the null distribution derived from DMSO controls. The red lines indicate significance thresholds without multiple testing correction ($P < 0.05$) and with Bonferroni correction (1772 compounds). FDR correction with the Benjamini–Hochberg method was applied to the data. The Bonferroni threshold is shown as this is the initial, strictest threshold applied in this method. (I) Two biological replicates were recorded for each xenobiotic-species pair. The Pearson correlation between these replicates is indicated in the heatmap. (J, K) Representative plots illustrating the correlation between the two biological replicates for different metabolite-species pairs.

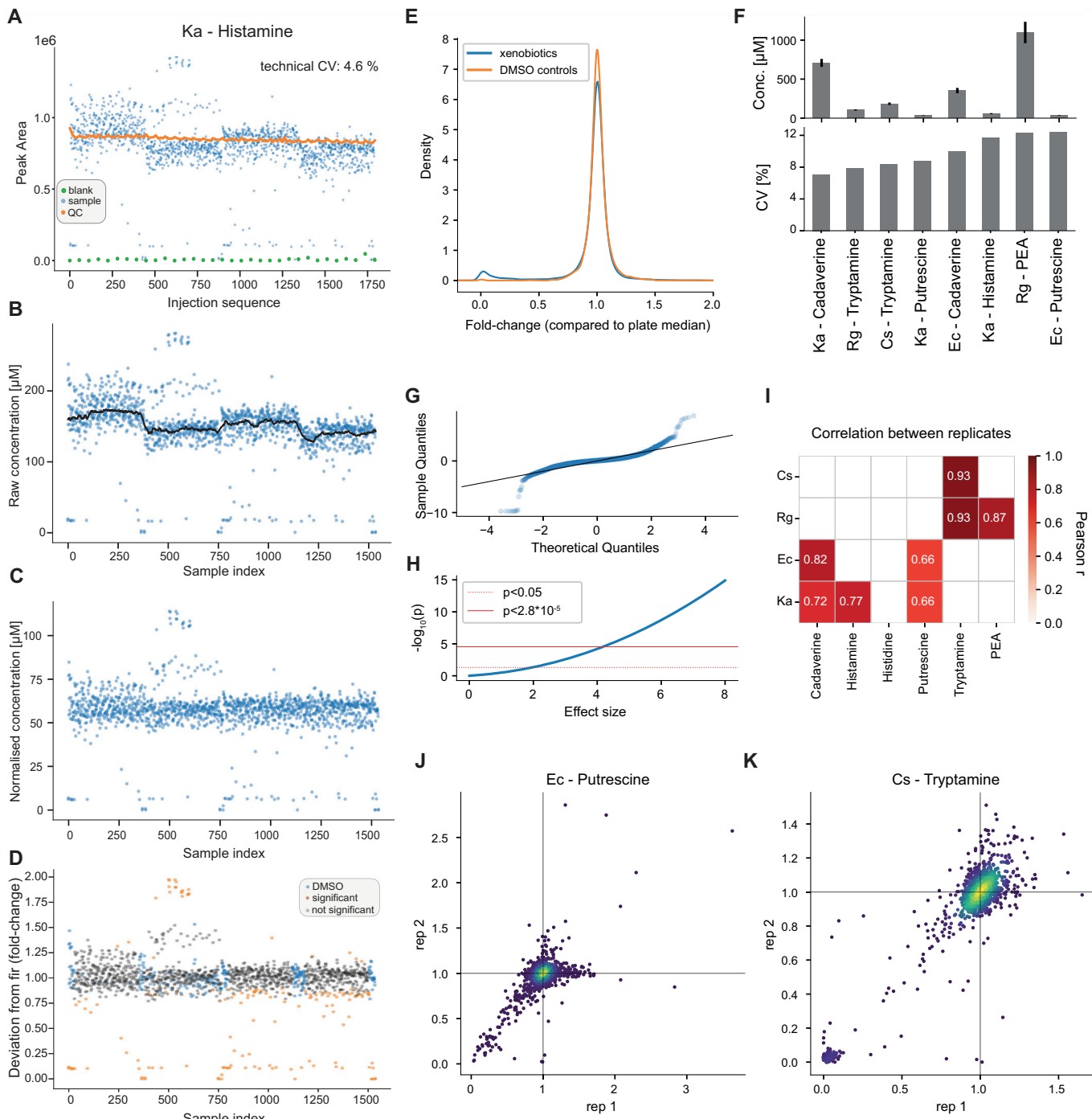

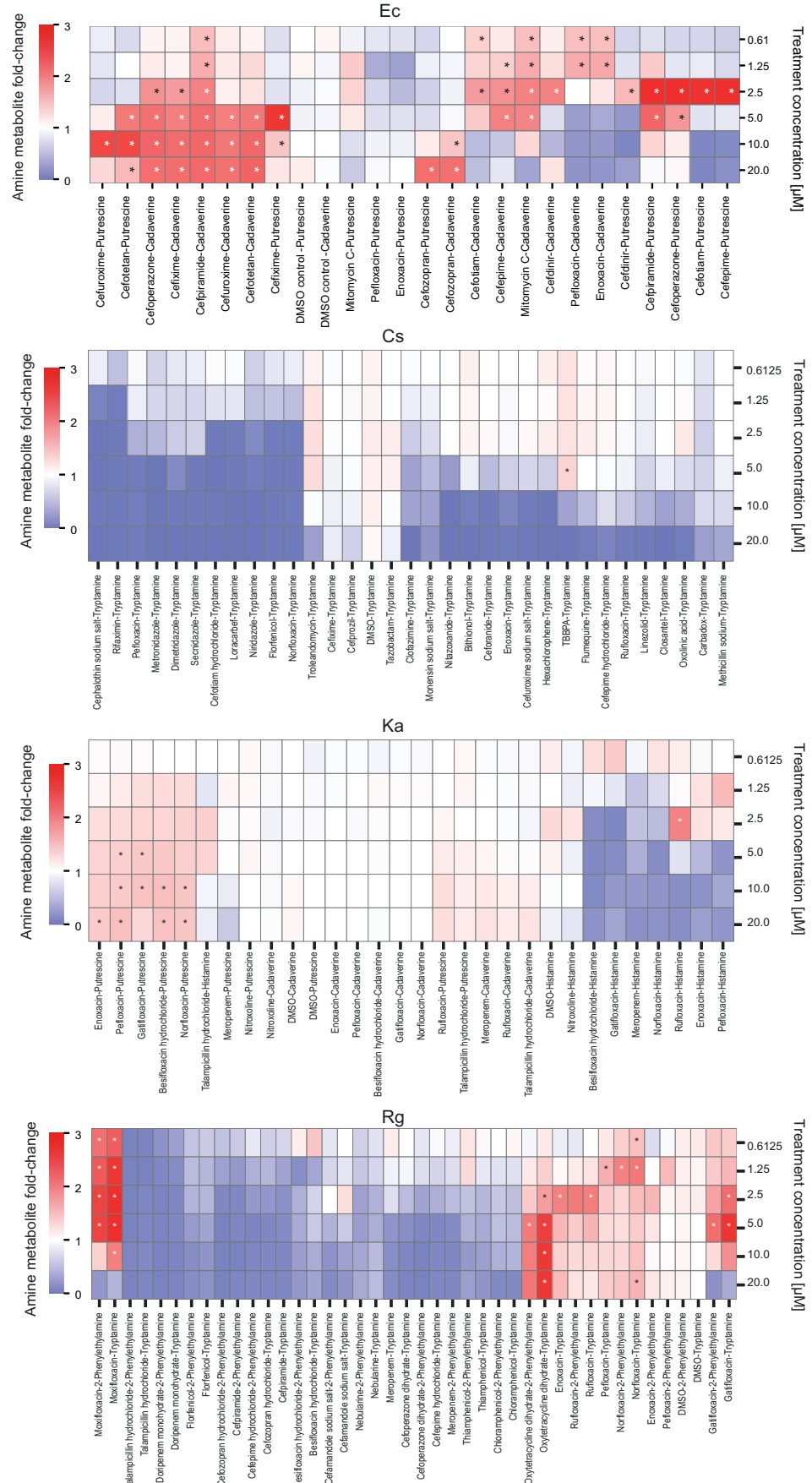

**Figure EV2.   Dose–response behaviour for selected xenobiotics.**

Heatmaps show amine metabolite fold changes across species (panels), compound-metabolite pairs (rows) and treatment concentrations (columns). Hits (same criteria as main screen: mean $\log_2$(fold change) >0.32, both replicates $P_{adj} < 0.05$ (two-sided FDR-corrected z-test), consistent direction of metabolite change) with increased amine production are marked with '*'. Heatmaps columns were clustered using Euclidean distance and the Average method.

                                                                                                  

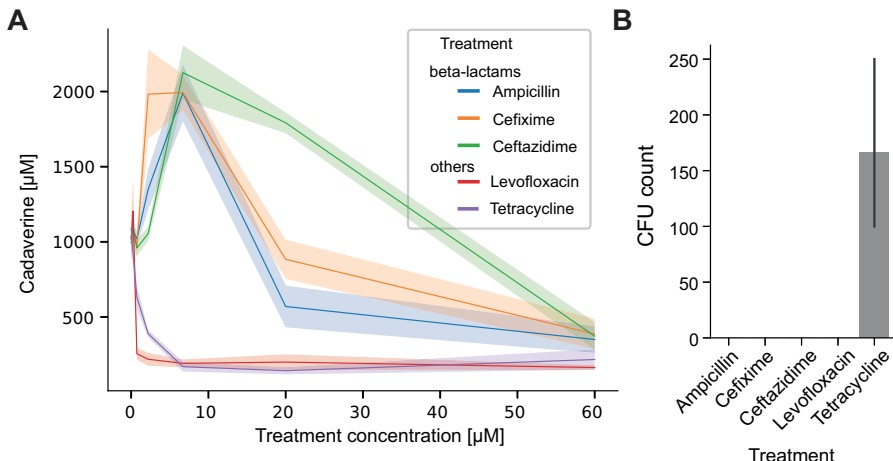

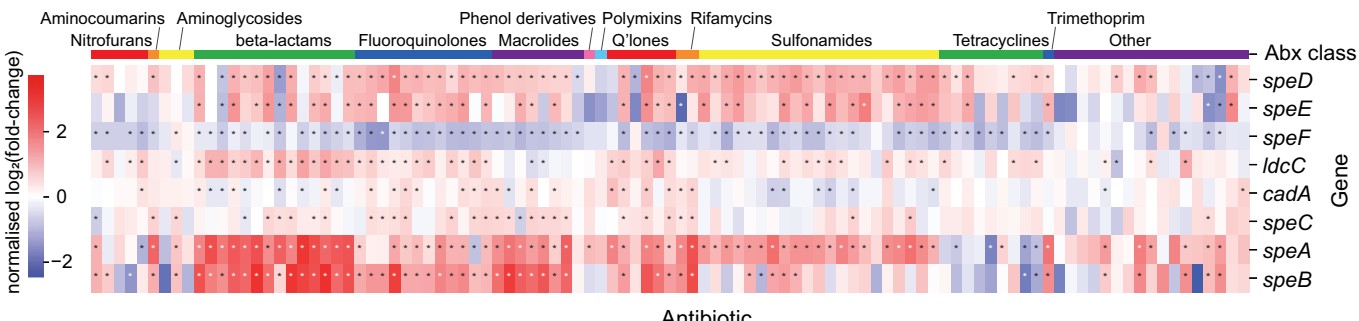

**Figure EV3.  Additional antibiotics experiments.**

(A) Validation of the main screen results using a subset of antibiotics from different classes, prepared and measured independently. Lines indicate the mean and shaded areas the standard deviation of $n = 3$ biological replicates. (B) Number of colony-forming units of *E. coli* after 3.5 h incubation in mGAM medium with 20 μM of each antibiotic. Bar heights indicate the mean and error bars the standard deviation of $n = 3$ biological replicates. (C) Analysis of published dataset by Guillen et al (Noto Guillen et al, 2024) capturing the fitness of pooled, genome-wide knock-out mutants in the presence of various drugs. The analysis is based on Supplementary Table 3 in the above publication, using the normalised log₂(fold-change) data (sheet 'normLFC') and adjusted $P$ values (sheet '$P_{adj}$', see the source publication for details). We filtered the data for known polyamine biosynthetic enzymes ($y$ axis) and antibiotics (column antibiotic == 1). Significant interactions are marked '*', using the same cut-off of $P_{adj} < 0.25$ as used by the authors. Rows were clustered using Euclidean distance and the Average method. Columns were sorted by antibiotic class.

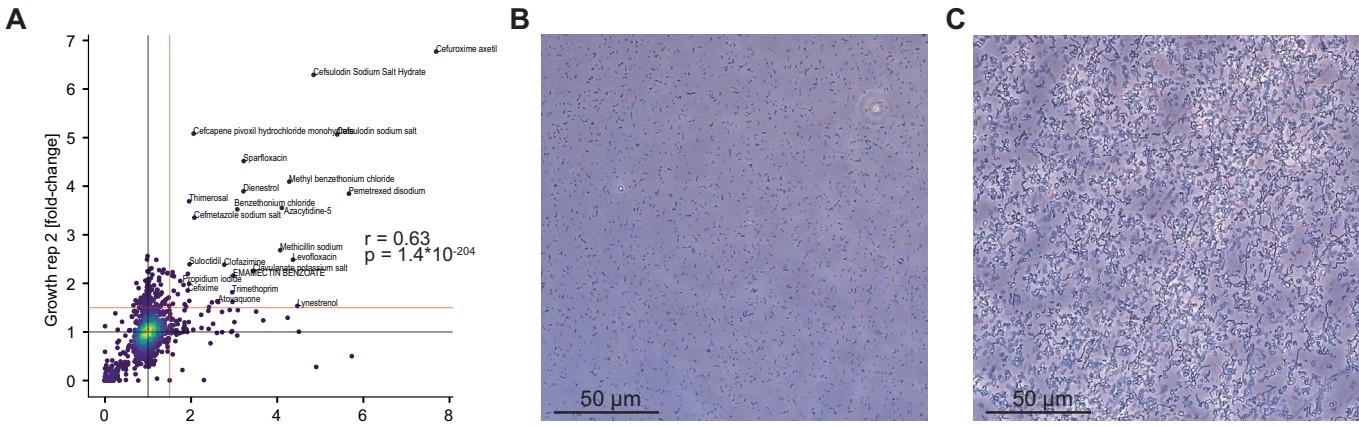

**Figure EV4.  Cell aggregation in *R. gnavus* confounds OD measurements.**

(**A**) Growth fold changes for *R. gnavus* across the two biological replicates. The Pearson correlation coefficient and associated two-sided *P* value is shown in the plot. Very high values are often consistent across replicates. (**B**) Phase contrast micrograph of *R. gnavus* cells in control conditions (mGAM 1% DMSO). Images were acquired using a Leica DM1000 LED microscope, a 20x/0.4 PH1 objective and Leica ICC50 W camera. (**C**) Similar micrograph of *R. gnavus* cells treated with compound causing abnormally high OD readings.

