## [Peer Review File · Molecular Systems Biology]

Impact of drugs and environmental contaminants on amine production by gut bacteria

Stephan Kamrad, Tara Davis, and Kiran Patil

Corresponding author(s): Kiran Patil (kp533@cam.ac.uk)

Review Timeline:

Submission Date:	27th Dec 24
Editorial Decision:	10th Feb 25
Revision Received:	2nd May 25
Editorial Decision:	2nd Jun 25
Revision Received:	11th Jun 25
Accepted:	16th Jun 25

Editor: Poonam Bheda

Transaction Report:

10th Feb 2025

Manuscript Number: MSB-2024-12833

Title: Impact of drugs and environmental contaminants on amine production by gut bacteria

Dear Prof Patil,

Thank you for the submission of your manuscript to Molecular Systems Biology. We have now received feedback from the three reviewers who agreed to evaluate your manuscript. As you will see from the reports below, the referees acknowledge the interest of the study and are overall supportive of your work; however they also comment on multiple aspects of the manuscript that should be strengthened in a revision.

Without repeating all the comments listed below, some of the more fundamental issues raised are the following:

- some mechanistic insight or clarifications for the specific amine increases should be included and would significantly enhance the study, as commented by all three reviewers
- additional and more appropriate statistical tests should be used to confirm correlations and conclusions, in line with comments from Reviewer 3
-

All other issues raised would need to be satisfactorily addressed. Please let me know in case you would like to discuss in further detail any of the comments, I would be happy to schedule a call.

We require:

- 1) A .docx formatted version of the manuscript text (including legends for main figures, EV figures and tables). Please make sure that the changes are highlighted to be clearly visible. Alternatively you may choose to submit your manuscript as a LaTeX file.
- 2) Individual production quality figure files as .eps, .tif, .jpg (one file per figure). For guidance, download the 'Figure Guide PDF' (<https://www.embopress.org/page/journal/17574684/authorguide#figureformat>).
- 3) At EMBO Press we ask authors to provide source data for the main figures. Our source data coordinator will contact you to discuss which figure panels we would need source data for and will also provide you with helpful tips on how to upload and organize the files.
- 4) A .docx formatted letter INCLUDING the reviewers' reports and your detailed point-by-point responses to their comments. As part of the EMBO Press transparent editorial process, the point-by-point response is part of the Peer Review File (PRF), which will be published alongside your paper.
- 5) A complete author checklist, which you can download from our author guidelines (<https://www.embopress.org/page/journal/17574684/authorguide#submissionofrevisions>). Please insert information in the checklist that is also reflected in the manuscript. The completed author checklist will also be part of the PRF.
- 6) Please note that all corresponding authors are required to supply an ORCID ID for their name upon submission of a revised manuscript.
- 7) It is mandatory to include a 'Data Availability' section after the Materials and Methods. Before submitting your revision, primary datasets produced in this study need to be deposited in an appropriate public database, and the accession numbers and database listed under 'Data Availability'. Please remember to provide a reviewer password if the datasets are not yet public (see <https://www.embopress.org/page/journal/17574684/authorguide#dataavailability>).

In case you have no data that requires deposition in a public database, please state so in this section as follows: "This study includes no data deposited in external repositories". Note that the Data Availability Section is restricted to new primary data that are part of this study.

8) All Materials and Methods need to be described in the main text using our 'Structured Methods' format, which is required for all research articles. According to this format, the Methods section includes a Reagents and Tools Table (listing key reagents, experimental models, software and relevant equipment and including their sources and relevant identifiers) followed by a Methods and Protocols section describing the methods using a step-by-step protocol format. The aim is to facilitate adoption of the methodologies across labs. Please upload the Reagents and Tools table as a separate document when submitting your

revised manuscript. More information on how to adhere to this format as well as a downloadable template (.docx) for the Reagents and Tools Table can be found in our author guidelines:

<https://www.embopress.org/page/journal/17444292/authorguide#structuredmethods>

9) For data quantification: please specify the name of the statistical test used to generate error bars and p-values, the number (n) of independent experiments (specify technical or biological replicates) underlying each data point and the test used to calculate p-values in each figure legend. The figure legends should contain a basic description of n, p-values and the test applied. Graphs must include a description of the bars and the error bars (s.d., s.e.m.). Please provide exact p-values (in either the figure or figure legend).

10) Our journal encourages inclusion of *data citations in the reference list* to directly cite datasets that were re-used and obtained from public databases. Data citations in the article text are distinct from normal bibliographical citations and should directly link to the database records from which the data can be accessed. In the main text, data citations are formatted as follows: "Data ref: Smith et al, 2001" or "Data ref: NCBI Sequence Read Archive PRJNA342805, 2017". In the Reference list, data citations must be labeled with "[DATASET]". A data reference must provide the database name, accession number/identifiers and a resolvable link to the landing page from which the data can be accessed at the end of the reference. Further instructions are available at .

11) We replaced Supplementary Information with Expanded View (EV) Figures and Tables that are collapsible/expandable online. EV Figures should be cited as 'Figure EV1, Figure EV2' etc... in the text and their respective legends should be included in the main text after the legends of regular figures.

- Additional Tables/Datasets should be labeled and referred to as Table EV1, Dataset EV1, etc. Legends should be provided in a separate tab in case of .xls files. Alternatively, the legend can be supplied as a separate text file (README) and zipped together with the Table/Dataset file.

<https://www.embopress.org/page/journal/17574684/authorguide#expandedview>

12) Author contributions: CRedit has replaced the traditional author contributions section because it offers a systematic machine-readable author contributions format that allows for more effective research assessment. Please remove the Authors Contributions from the manuscript and use the free text boxes beneath each contributing author's name in our system to add specific details on the author's contribution. More information is available in our guide to authors.

13) Disclosure statement and competing interests: We updated our journal's competing interests policy in January 2022 and request authors to consider both actual and perceived competing interests. Please review the policy

<https://www.embopress.org/competing-interests> and update your competing interests if necessary.

14) Every published paper now includes a 'Synopsis' to further enhance discoverability. Synopses are displayed on the journal webpage and are freely accessible to all readers. They include a short stand first (maximum of 300 characters, including space) as well as 2-5 one-sentences bullet points that summarizes the paper. Please write the bullet points to summarize the key NEW findings. They should be designed to be complementary to the abstract - i.e. not repeat the same text. We encourage inclusion of key acronyms and quantitative information (maximum of 30 words / bullet point). Please use the passive voice. Please attach these in a separate file or send them by email, we will incorporate them accordingly.

Please note that these would be the final versions and changes during proofing are usually not allowed.

15) As part of the EMBO Publications transparent editorial process initiative (see our policy here:

https://www.embopress.org/transparent-process#Review_Process), Molecular Systems Biology will publish online a Peer Review File (PRF) to accompany accepted manuscripts.

In the event of acceptance, this file will be published in conjunction with your paper and will include the anonymous referee reports, your point-by-point response and all pertinent correspondence relating to the manuscript. Let us know whether you agree with the publication of the PRF and as here, if you want to remove or not any figures from it prior to publication.

Please note that the Author checklist will be published at the end of the PRF.

Molecular Systems Biology has a "scooping protection" policy, whereby similar findings that are published by others during review or revision are not a criterion for rejection. Should you decide to submit a revised version, I do ask that you get in touch after three months if you have not completed it, to update us on the status.

I look forward to receiving your revised manuscript.

Yours sincerely,

Poonam Bheda, PhD
Scientific Editor
Molecular Systems Biology

Reviewer #1:

In this short report, Kamrad et al address the effect of several xenobiotics (including common drugs and sweeteners) on overall biogenic amine production by several gut bacteria. Their general (unsurprising) finding is that many xenobiotics will disrupt metabolic homeostasis in these bacteria. Of interest though is the finding that several of these can disrupt amine production/metabolism in a growth independent manner - and that sometimes low doses of xenobiotics have very strong effects on metabolic outputs.

This is an interesting and impressively well done short study - and has several useful findings, although not all findings are surprising. In general this is also a very useful resource - given the scale at which this study has been done - to understand how xenobiotics alter microbial metabolite production and microbe-microbe interactions in the microbiome (and goes beyond just their effect on growth).

I have several points - which can be addressed by either experiment or nuanced clarifications.

1) For better clarity of some of the metabolite data, consider including a small supplemental selection showing replicates for amine measurement, for a few key xenobiotics (for the no-growth change/change in amine, or growth-change/change in amine). eg related to Fig 2

2) The paper does not try to address any possible common reason for specific amine increases - and whether it comes from a protective-stress response (eg rewiring Arg metabolism towards polyamines), or is a more direct mechanism. Any clarifications/insights in this regard will enhance the manuscript (beyond the very interesting observations).

3) Additionally - with respect to some xenobiotics that cause polyamine increase - this is similar to that seen with some (especially anti-translation inhibitor) antibiotics at sub IC50. A small comparison to see if these trends are similar would be useful. Also, the b-lactams vs the translation inhibitors would have slightly different effects - a small comparison along these lines (expanding on figure 4, and supplementary Fig 3B) would be very useful.

4) The growth outcomes (and lack of effect on growth despite amine increase) is interesting. However, growth (in terms of biomass) is a late end point outcome. A better read-out of changes in 'growth state' would be changes in ribosomal biogenesis. There are older studies (mid 2000s, and more recent ones on allocations of arginine towards either ribosome biogenesis or polyamine metabolism) in e. coli and yeast that suggest a general trend of somewhat inverse correlation between increased polyamine synthesis and decreased translation and/or ribosomal biogenesis. A useful addition (to the growth data) with xenobiotics would be to see if there is any effect on ribosomal biogenesis (at the level of ribosomal transcripts) for some of the xenobiotics that do not show any change in growth (end point biomass), but have substantial amine production.

5) How much monoamines are produced? While there is a clear increase - in terms of absolute concentrations, how much is produced, and is it of substantial amounts - since this can be important for gut function and/or other functions. If there is an increase, but to amounts that cannot be substantial, then this finding is real, but may be inconsequential.

minor points:

1) Intro/discussion - Among summarising biogenic amine roles - polyamides have multiple, more important roles in protecting cells against osmotic, desiccation stress as well as nucleic acid damage (beyond just oxidative stress). Worth a mention, given that the xenobiotics might trigger these responses.

2) Again minor clarification point - bacteria produce a larger range of, and higher amounts of biogenic amines than mammals.

Reviewer #2:

Kamrad et al have prepared a manuscript on the impact of drugs, using a library of >1700 compounds, on biogenic amine production by gut bacteria.

The study clearly includes a lot of data and integration of different data types: microbiology, metabolomics and data analysis, reported in a fairly solid way, with attractive figures.

Fundamentally a main question that is not addressed in this study that should be more explored, especially in the Results and Discussion: Why do some drugs stimulate biogenic amine production, and especially: what is the mechanism? Why do these biogenic amine stimulate growth...? Do they function as carbon source, and if so, how, etc? This interpretation would be great to include, as otherwise the study remains quite explorative, but also descriptive. The cause for biogenic amine modulation is not clear.

While the manuscript seems quite solid in data and reporting, the Figures while attractive are sometimes confusing, or seem to display the same data in different ways, which dilute the message.

Specific comments:

- Figure 1A: in all structures, it is conventional to show the amine group as -NH₂, by explicitly mentioning the hydrogens, and not -N, and shown; and the same applies to the -COOH group (instead of -COO).

Example how one should represent histidine:

Figure 1C: this graph is difficult to interpret - 16% of the drugs affected the concentration of at least 1 amine. What is the 17%? And what are the black and red blocks below?

- Figure 2 - legend B is cryptic, please reformulate

- Figure 2C - the legend is black and white but the figure has 3 colors: pink, red and green... please rethink the legend or the figure

- Figure 2D - what is PEA?

- P13 - Please try to be consistent when reporting units: either mL or ml

- P13, L25 - avoid reporting rpm as it depends on the size of the centrifuge's rotor

- P14, L3 - 'cold extraction buffer' implies a buffered solution, that is not the case of what is reported to have been used, as there isn't any buffer capacity in a solution of acetonitrile, methanol, formic acid and two other compounds....

- P14, L9-22 - the details of LC-MS methods: settings, column, MRM transitions per compound, etc need to be reported...

- P5, L22 - here, one expects reporting of some quantitative values on replicate variability, and what is considered as "high data consistency". What was a minimum and maximum variation, etc? Were some of the replicates excluded due to too high variability - what were the criteria? Given that the authors performed a very large screen and even were able to correct batch effects throughout measurements, as depicted in Fig1S, it would be good to report these "errors".

Reviewer #3:

Summary

- Describe your understanding of the story

In this work, the authors implement a high-throughput targeted metabolomics assay to measure amine production of gut bacteria in response to xenobiotics, including pharmaceuticals (antibiotics), pesticides, and other clinically and agriculturally relevant compounds. Leveraging growth data, they compare the relationship between growth rate and amine production in response to different xenobiotics and reveal an interesting, but perhaps not surprising finding, that the production of amines does not necessarily correlate with the magnitude or direction of the impact on growth. These results suggest that the bacteria are executing specific responses to these xenobiotics, and the high degree of responses suggest a greater complexity to this response. The authors go on to show that antibiotics trigger polyamine production in *E. coli*. The review process lends itself to a more critical tone, so I would like to emphasize that the study is valuable and just needs more appropriate statistics and deeper insights to highlight all it has to offer.

- What are the key conclusions: specific findings and concepts

1. Decoupling of growth rate and amine production. In some cases (the proportion varies across species), amine production is significantly altered but not proportional to growth changes. This indicates a more compound-specific stress response is occurring (usually at low doses of the xenobiotic).

2. Amine production follows non-linear response curve to Abx. As xenobiotic dosage increases, there appears to be a burst of amine production at low concentrations that decreases at increased concentrations.

3. Growth responses between related species are more consistent whereas the amine production varies. *E. coli* and *K. aerogenes* show similar growth responses to xenobiotics but *E. coli* overproduces polyamines, whereas *K. aerogenes* does not. This was

- What were the methodology and model system used in this study

The authors implement an impressive high-throughput targeted metabolomics assay to measure amine production (N=6) of four clinically-relevant gut bacteria in response to xenobiotic exposure. They also perform dose-response curves, measuring growth rate and amine production across a range of xenobiotics for some specific cases.

General remarks

- Are you convinced of the key conclusions?

1. Decoupling of growth rate and amine production. The data suggests this is largely true, however, the correlation analyses leading to these conclusion are flawed in their reliance on pearson correlations, which are heavily skewed by outlier data points. Furthermore, a statistical test (e.g. Fishers z-test) between these correlation values can be performed to establish their significance.

2. Amine production follows non-linear response curve to Abx. This is quite clear from figures 3E-G however, may not be surprising. One interpretation of this is that while bacteria can execute these stress responses at low concentrations of xenobiotic, they become overwhelmed at higher concentrations and die out or opt for other response pathways.

3. Growth responses between related species are more consistent whereas the amine production varies. This is evident in figure 3A/B. While I expect these results will hold, they are once again using pearson correlation as the primary metric and lack a definitive statistical test to compare the correlations.

- Place the work in its context.

In recent years, there have been several studies on drug-microbe interactions and it is known that these screens lead to the discovery of novel mechanisms with potentially clinical implications. This is the largest I have seen that combines both growth rate and amine production across multiple clinically-relevant species.

- What is the nature of the advance (conceptual, technical, clinical)?

This advance is primarily technical and really pushes the scale by which we can collect multidimensional data, especially amine production.

- How significant is the advance compared to previous knowledge?

The authors highlight that this is the first pan-xenobiotic scale overview of the relationship. There is a great degree of new, useful, data that could lead to novel hypotheses about drug-microbe-host interactions and new biological mechanisms. While the potential is great, nothing explored or presented does not strike me as a very significant jump. There are many small vignettes and insights drawn in this study but the authors could do better to highlight or emphasize the more impactful findings.

- What audience will be interested in this study?

This study will be of interest to clinicians interest in integrative health solutions that consider the role of the microbiome in drug responses. Microbiologists interested in the gut microbiome will be interested in how amine production in response to low-dose xenobiotics may influence microbial interactions and the overall ecology of the microbiome. There may also be implications for environmental bacteria that may be exposed to low doses of xenobiotics, which would be interesting.

Major points

-Specific criticisms related to key conclusions

1) Coupling vs decoupling of metabolic responses to growth rate is largely determined using pearson correlations, which are highly sensitive to outliers. Given that the distribution of responses are biased, with mostly minimal changes and a few extremely strong responses, more robust correlation metrics should be applied (Biweight Mid Correlation (Bicor), Spearman). You may also consider first filtering for only drug-microbe combinations with significant differences in metabolite production before assessing correlations, which would reduce the bias in your distributions. Furthermore, correlations should be reported with p-values. Also, if you are claiming one correlation is stronger than another, a statistical test (Fishers z-score perhaps?) should be performed to support this claim.

2) There are many hits it seems yet in figure 4 the authors choose to focus on the least interesting, ones with known antibiotic effects that would obviously trigger stress responses. Perhaps the story could be reworked to motivate this instead. Why is it important to know that low-doses of antibiotics activate amine production? I can imagine clinical implications, but these types of rationales for what you investigate deeper should be clearly stated.

The impact of this paper hinges on the consequences of the amine production response. That there is a stress response to antibacterials is not very exciting (barring sufficient rationale). Figure 4, focusing on antibacterials would be much more interesting if included a figure like 4C, but with a xenobiotic that is not explicitly antibacterial. Do any of the antidepressants or

chemotherapeutics stimulate amine production? How might this amine production influence patient outcomes?

3) There is a lot of emphasis on the non-monotonous dose dependency of growth and amine production, but it seems like there is a stress response at low doses (as expected by antibiotics), but the cells just die at higher doses. Why is this important or what insights does this give us?

4) The argument that metabolite production is coupled to growth seems to overlook the fact that fewer cells will produce less metabolites. I did not see a description in the methods of how measured production would be normalized to absolute growth rate or biomass. As is, it suggests that in these cases the cells adjust production in response to xenobiotic exposure in an amount that is commensurate to the effect on growth rate. Another interpretation is that, there are just fewer cells so production is lower.

-Specify experiments or analyses required to demonstrate the conclusions

1) Recalculate correlations with more robust correlation metrics and appropriate statistical tests

2) I would really like to see 1) More examples of surprising or new insights or 2) mechanisms of bacterial responses in this work. As is, it's clear a lot of data was generated but I don't think any especially surprising or interesting results were reported. I am sure they are there. Either of these two analyses would cement the value of the data generated in this work and how it influences clinical work or improves our understanding of microbial responses to xenobiotics.

For surprising results:

1) More investigations into amine production in response to non-antibacterials. The authors could highlight drugs that are not explicitly antibacterials, but still elicit amine production. Additional discourse on how these amines might modulate host physiology or clinical outcomes would be a huge plus. You have already identified many xenobiotics that stimulate metabolite production, but don't affect growth (2F), so this should be relatively straightforward.

For mechanisms:

2) Are there different pathways regulating production of these amines? What regulatory programs are controlling these responses and are they shared between different classes of xenobiotics? A few examples of RNA-Seq in response to xenobiotic exposure or RT-qPCR comparing these responses between different classes of xenobiotics would be nice to see. Perhaps other studies such as (<https://doi.org/10.1038/s41564-023-01581-x>) can provide this data.

-Motivate your critique with relevant citations and argumentation

As highlighted in your introduction, stress responses triggers polyamine production in several species. The paper mentioned above (<https://doi.org/10.1038/s41564-023-01581-x>) is a nice demonstration of high-throughput screens with follow up studies to establish new mechanistic insights. Granted, transcriptional data is more suited for this, but RNA-seq in E. coli is a fairly straightforward pipeline that could be applied here.

Minor points

-Easily addressable points

1) It would be nice to have a table describing the amines you measure, and perhaps a brief description of their clinical relevance.

2) Correlations should include p-values

3) Page 10, line 2. Figure 4 is about cadaverine, it would be nice to have a rationale for why this is important. With all the data, why focus on this story?

4) Page 10, Line 16: You highlight bacteriostatic antibiotics despite not having data for any. Unless you have data for bacteriostatic abx, I would suggest removing this as it sets up expectations. Line 18 describes how you found no link since all B-lactams were cidal, which is a poor statement to make since you didn't test any bacteriostatic compounds. Bacteriostatic vs -cidal and amine production would have been really interesting though.

5) Page 13, line 41: What did you do with the data in cases where the OD was confounded by the aggregation?

6) Page 11 line 23-25: I don't really understand the latter part of this sentence.

-Presentation and style

1) 2C: Instead of dark and light colors, striped and solid could be more clear.

2) 2G is a bit hard to read. You could add a gap in between the columns between different amines.

-Trivial mistakes

1) 3B and 3C are not mentioned in the text

2) 3G is not mentioned in the text or the legend, despite being an important figure and quite possibly one of the highlights of this work.

3) 3E/F Should indicate that this is R. gnavus in the legend or figure.

4) Page 12, line 6: Italicize and correct spelling for Enterobacteriaceae

5) Page 14, line 24: Python version is incorrect here

For major revision, it is useful if you can provide a time estimation for the requested additional experiments/analyses.

1. Reanalyze with appropriate correlation metrics and statistical significance test (3 days)
2. More investigations into amine production in response to non-antibacterials. If you could show something like figure 4C but for a non antibacterial (especially a drug), that would be very interesting. (1 month)

Or

3. RNA-seq of *E. coli* in response to a few non-antibacterial xenobiotics to map stress response to generate polyamines (1-2 month). Looking at correlations between transcription profiles for different xenobiotics would be interesting to see whether or not they are similar or different.

Summary of main changes

We thank all reviewers for their supportive and constructive feedback and suggestions. In response, we have conducted additional experiments, analyzed related, independent, publicly available datasets, and refined the statistical analysis and presentation of the data.

1. To gain insights into the mechanisms regulating amine production, we have:

- Analyzed a large-scale transcriptomics study by Ricaurte et al (2024) to investigate the mechanism of increased cadaverine production by the antidepressant paroxetine.
- Analyzed a published genome-wide pooled knock-out mutant screen by Guillen et al (2024) to investigate the fitness of polyamine biosynthesis mutants.
- Expanded the Discussion of potential mechanisms, with a focus on anaerobic conditions.

2. Experimentally, we have conducted additional follow-ups for three non-antibiotics (the HIV drug didanosine, the antidepressant paroxetine and the pesticide fenazaflor), further highlighting the impact of diverse non-antibiotics on amine metabolism.

3. Statistical analyses have been improved, particularly regarding the use of different correlation metrics and reporting of p-values.

Editorially requested changes

We require:

We include a docx file of the revised manuscript and a version with tracked changes.

2) Individual production quality figure files as .eps, .tif, .jpg (one file per figure). For guidance, download the 'Figure Guide PDF'

Individual figures are provided as eps files (in addition to being embedded in the manuscript for the convenience of the reviewers).

We have been in touch with Source Data Editor, Dr. Hannah Sonntag, and they advised us that the data currently provided is sufficient.

5) A complete author checklist

This is now provided as part of the submission.

7) It is mandatory to include a 'Data Availability' section after the Materials and Methods.

Data availability statement is provided.

8) All Materials and Methods need to be described in the main text using our 'Structured Methods' format.

We have adapted the Methods section accordingly and now include a reagents table and stepwise protocols.

9) For data quantification: please specify the name of the statistical test used to generate error bars and p-values, the number (n) of independent experiments (specify technical or biological replicates) underlying each data point and the test used to calculate p-values in each figure legend. The figure legends should contain a basic description of n, p-values and the test applied. Graphs must include a description of the bars and the error bars (s.d., s.e.m.). Please provide exact p-values (in either the figure or figure legend).

This information is provided.

10) Our journal encourages inclusion of *data citations in the reference list* to directly cite datasets that were re-used and obtained from public databases.

We have not used any public databases.

11) We replaced Supplementary Information with Expanded View (EV) Figures and Tables that are collapsible/expandable online. EV Figures should be cited as 'Figure EV1, Figure EV2" etc... in the text and their respective legends should be included in the main text after the legends of regular figures.

We have renamed the figures and datasets accordingly.

12) Author contributions: CRediT has replaced the traditional author contributions section because it offers a systematic machine-readable author contributions format that allows for more effective research assessment. Please remove the Authors Contributions from the manuscript

and use the free text boxes beneath each contributing author's name in our system to add specific details on the author's contribution. More information is available in our guide to authors.

Author contributions have been removed from the main manuscript.

13) Disclosure statement and competing interests: We updated our journal's competing interests policy in January 2022 and request authors to consider both actual and perceived competing interests. Please review the policy <https://www.embopress.org/competing-interests> and update your competing interests if necessary.

We have reviewed the policy and declare no conflicts of interest.

14) Every published paper now includes a 'Synopsis' to further enhance discoverability. Synopses are displayed on the journal webpage and are freely accessible to all readers. They include a short stand first (maximum of 300 characters, including space) as well as 2-5 one-sentences bullet points that summarizes the paper. Please write the bullet points to summarize the key NEW findings. They should be designed to be complementary to the abstract - i.e. not repeat the same text. We encourage inclusion of key acronyms and quantitative information (maximum of 30 words / bullet point). Please use the passive voice. Please attach these in a separate file or send them by email, we will incorporate them accordingly.

Please note that these would be the final versions and changes during proofing are usually not allowed.

We have added the synopsis and accompanying image to the title page and are also submitting these as separate files.

15) As part of the EMBO Publications transparent editorial process initiative

We appreciate this policy and have no objections to publishing the point-by-point responses in full.

Reviewer #1:

In this short report, Kamrad et al address the effect of several xenobiotics (including common drugs and sweeteners) on overall biogenic amine production by several gut bacteria. Their general (unsurprising) finding is that many xenobiotics will disrupt metabolic homeostasis in these bacteria. Of interest though is the finding that several of these can disrupt amine production/metabolism in

a growth independent manner - and that sometimes low doses of xenobiotics have very strong effects on metabolic outputs.

This is an interesting and impressively well done short study - and has several useful findings, although not all findings are surprising. In general this is also a very useful resource - given the scale at which this study has been done - to understand how xenobiotics alter microbial metabolite production and microbe-microbe interactions in the microbiome (and goes beyond just their effect on growth).

I have several points - which can be addressed by either experiment or nuanced clarifications.

We would like to thank the reviewer for their efforts in evaluating our manuscript and the constructive feedback. We have implemented the suggestions as outlined below and believe that this has substantially improved the manuscript.

R1.1 - Show replicates

For better clarity of some of the metabolite data, consider including a small supplemental selection showing replicates for amine measurement, for a few key xenobiotics (for the no-growth change/change in amine, or growth-change/change in amine). eg related to Fig 2

We have generated the requested figure below, showing individual replicates for all hit compounds, classified by growth concordance. However, we do not feel that the manuscript would benefit from including it as an additional supplement, as we already have detailed quality control and replicate correlation analyses (Figure EV1) and entirely independent validations/follow-ups for many compounds.

Response Figure 1: Plots show individual datapoints from the main screen, for hit compounds only, grouped by species and amine metabolite. The two biological replicates are connected with a line. The colour of the line indicates whether the hit was classified as growth concordant. Grey dotted lines show the null effect for orientation, red dotted lines indicate the fold-change cut-off applied to the mean amine abundance. For *R. gnavus*, a small number of OD values (but not metabolomics values) were excluded due to clumping, therefore some compounds only have one replicate in this plot.

R1.2 - Mechanisms

The paper does not try to address any possible common reason for specific amine increases - and whether it comes from a protective-stress response (e.g. rewiring Arg metabolism towards polyamines), or is a more direct mechanism. Any clarifications/insights in this regard will enhance the manuscript (beyond the very interesting observations).

Thank you for this comment. We have expanded the mechanistic aspects of the manuscript in three ways:

First, we have analyzed a publicly available dataset [Ricaurte et al (2024), *Nature Microbiology*] that describes the transcriptional responses of diverse gut bacteria to selected pharmaceutical drugs. For paroxetine, a selective serotonin reuptake inhibitor for which we observe an upregulation of cadaverine production without growth changes at 20 μM in *E. coli*, two cadaverine biosynthesis genes and an acid stress response protein are strongly upregulated. This is strong evidence for a transcriptional upregulation of cadaverine biosynthesis driven by *cadA* (rather than *ldcC*, the other lysine decarboxylase of *E. coli*) (new Figure 4G-H).

Second, we have re-analysed another publicly available dataset [Guillen et al (2024), *Science*] that captures the abundance of genome-wide knock-out mutants in the presence of numerous antibiotics and non-antibiotics. Focusing on the 8 key polyamine-producing enzymes [Chattopadhyay et al (2009), *J. Bacteriology*], we detect strong abundance changes of the corresponding knock-out strains in pooled cultures compared to untreated controls, however no clear pattern that distinguishes antibiotic classes or polyamine pathways is apparent. Surprisingly, mutants (especially *speA* and *speB*) for individual polyamine-biosynthesis genes often have significantly improved fitness in the mutant pool when challenged with diverse antibiotics, suggesting that these functions are not required for improved growth/survival or compensated by polyamine sharing within the pooled culture (new EV Figure 3C).

Finally, we have expanded and added a new section in the Discussion to better summarize the literature around this topic. How exactly antibiotics kill bacterial cells (downstream of their direct targets) is not clear, especially in anaerobic conditions. A small number of studies have linked antibiotics exposure to redox and acid stress responses, which in turn trigger polyamine metabolism, although only for a few compounds and species, and always in aerobic conditions.

R1.3 - Translation inhibitors and beta-lactams

Additionally - with respect to some xenobiotics that cause polyamine increase - this is similar to that seen with some (especially anti-translation inhibitor) antibiotics at sub IC50. A small comparison to see if these trends are similar would be useful. Also, the b-lactams vs the translation inhibitors would have slightly different effects - a small comparison along these lines (expanding on figure 4, and supplementary Fig 3B) would be very useful.

We agree that this is interesting. Out of the three main classes of antibiotics which inhibit translation, aminoglycosides and macrolides were largely ineffective in *E. coli*. Tetracyclines were generally effective and reduced growth and amine production concordantly (Fig 4B). Tetracycline was already included as a follow up and did not trigger polyamine production at any concentration tested. We have adapted the Results section to better describe these differences between antibiotic classes.

Further dissecting the effect of different antibiotics classes (annotation from [62]), we identified 24 β -lactam antibiotics, which inhibit biosynthesis of the cell wall, as the main stimulators of cadaverine metabolism (**Figure 4B**). Fluoroquinolones, disrupting DNA, and tetracyclines, disrupting translation, mostly showed strong, growth-concordant reductions in cadaverine production. Other antibiotic classes like macrolides and aminoglycosides (also both translation inhibitors), and sulphonamides (targeting folate metabolism) showed little effect on both metabolism and growth, consistent with natural and acquired resistance of *E. coli* to these compounds [63–65].

R1.4 - Ribosome biogenesis

The growth outcomes (and lack of effect on growth despite amine increase) is interesting. However, growth (in terms of biomass) is a late end point outcome. A better read-out of changes in 'growth state' would be changes in ribosomal biogenesis. There are older studies (mid 2000s, and more recent ones on allocations of arginine towards either ribosome biogenesis or polyamine metabolism) in *e. coli* and yeast that suggest a general trend of somewhat inverse correlation between increased polyamine synthesis and decreased translation and/or ribosomal biogenesis. A useful addition (to the growth data) with xenobiotics would be to see if there is any effect on ribosomal biogenesis (at the level of ribosomal transcripts) for some of the xenobiotics that do not show any change in growth (end point biomass), but have substantial amine production.

Thank you for pointing out this interesting link. We have added a sentence and references in the Introduction to point out the link between polyamines and translation. Furthermore, we have analyzed a publicly available transcriptomics dataset by Ricaurte et al [PMID: 38233648] and have found that ribosomal proteins were transcriptionally upregulated under paroxetine (antidepressant, increases amine production in our screen without affecting growth). While we do not disagree that further investigating the impact on translational activity could be interesting, this would not add much to the main conclusions of the paper and is thus out of scope for the current study.

R1.5 - Relevance of absolute concentrations

How much monoamines are produced? While there is a clear increase - in terms of absolute concentrations, how much is produced, and is it of substantial amounts - since this can be important for gut function and/or other functions. If there is an increase, but to amounts that cannot be substantial, then this finding is real, but may be inconsequential.

Thank you for raising this important point. We selected the metabolites and species because they exhibit robust and substantial (high micromolar range) baseline concentrations, and a clinical relevance of the metabolite and species was known from the literature. We have added a graph showing the concentrations produced in DMSO-treated controls (**Fig EV1F**). We now also mention the serum concentration of these amines in the human host and highlight that in particular aromatic amines have potent effects even at low concentrations (Results):

“Robust and substantial production of amines was observed for all amine-bacteria pairs at baseline conditions (DMSO-treated control cultures), with concentrations ranging from 35 to 1100 μ M (Figure EV1F). This production capacity is significant in the context of the nanomolar-range physiological concentrations of these amines in the human body [53–56] and the ability of aromatic trace amines to exert strong effects at very low concentrations [9,57].”

R1 - minor points

1) Intro/discussion - Among summarising biogenic amine roles - polyamides have multiple, more important roles in protecting cells against osmotic, desiccation stress as well as nucleic acid damage (beyond just oxidative stress). Worth a mention, given that the xenobiotics might trigger these responses.

Thank you for raising this point. We have extended the Introduction in this regard:

“[27–29]. Polyamines also protect cells against other stresses, such as oxidative stress [30,31], osmotic stress [32] and DNA damage [33–35].”

2) Again minor clarification point - bacteria produce a larger range of, and higher amounts of biogenic amines than mammals.

Thank you for pointing this out. We have added a reference exploring the diversity of amine metabolism (Minguet et al) in the Introduction and now also describe the absolute concentration of amines produced.

Reviewer #2:

Kamrad et al have prepared a manuscript on the impact of drugs, using a library of >1700 compounds, on biogenic amine production by gut bacteria. The study clearly includes a lot of data and integration of different data types: microbiology, metabolomics and data analysis, reported in a fairly solid way, with attractive figures.

We thank the reviewer for the constructive feedback. We have improved the manuscript in response to the points raised.

Fundamentally a main question that is not addressed in this study that should be more explored, especially in the Results and Discussion: Why do some drugs stimulate biogenic amine production, and especially: what is the mechanism? Why do these biogenic amine stimulate growth...? Do they function as carbon source, and if so, how, etc? This interpretation would be great to include, as otherwise the study remains quite explorative, but also descriptive. The cause for biogenic amine modulation is not clear.

We agree with the point raised here, which is also shared by the other reviewers. We have expanded the mechanistic aspects of the manuscript in three ways:

First, we have analyzed a publicly available dataset [Ricaurte et al (2024), *Nature Microbiology*] that describes the transcriptional responses of diverse gut bacteria to selected pharmaceutical drugs. For paroxetine, a selective serotonin reuptake inhibitor for which we observe an upregulation of cadaverine production without growth changes at 20 μM in *E. coli*, two cadaverine biosynthesis genes and an acid stress response protein are strongly upregulated. This is strong evidence for a transcriptional upregulation of cadaverine biosynthesis driven by *cadA* (rather than *ldcC*, the other lysine decarboxylase of *E. coli*) (new Figure 4G-H).

Second, we have re-analyzed another publicly available dataset [Guillen et al (2024), *Science*] that captures the abundance of genome-wide knock-out mutants in the presence of numerous antibiotics and non-antibiotics. Focusing on the 8 key polyamine-producing enzymes [Chattopadhyay et al (2009), *J. Bacteriology*], we detect strong abundance changes of the corresponding knock-out strains in pooled cultures compared to untreated controls, however no clear pattern that distinguishes antibiotic classes or polyamine pathways is apparent. Surprisingly, mutants (especially *speA* and *speB*) for individual polyamine-biosynthesis genes often have significantly improved fitness in the mutant pool when challenged with diverse antibiotics, suggesting that these functions are not required for improved growth/survival or compensated by polyamine sharing within the pooled culture (new EV Figure 3C).

Finally, we have expanded and added a new section in the Discussion to better summarize the literature around this topic. How exactly antibiotics kill bacterial cells (downstream of their direct targets) is not clear, especially in anaerobic conditions. A small number of studies have linked antibiotics exposure to redox and acid stress responses, which in turn trigger polyamine metabolism, although only for a few compounds and species, and always in aerobic conditions.

While the manuscript seems quite solid in data and reporting, the Figures while attractive are sometimes confusing, or seem to display the same data in different ways, which dilute the message.

We have improved the clarity and presentation of the figures. We agree that there was some duplication in Figure 3 and have moved the old Figure 3G (large heatmap showing all dose-response results for *E. coli*) into the Expanded View figure.

R2 - Minor points

Figure 1A: in all structures, it is conventional to show the amine group as -NH₂, by explicitly mentioning the hydrogens, and not -N, and shown; and the same applies to the -COOH group (instead of -COO).

Thank you for the suggestion. We had drawn the chemical structures with the Pubchem sketcher with default settings which omits all hydrogens. But we agree that it is conventional and useful to show these hydrogens and have redrawn the structures in ChemDraw.

Figure 1C: this graph is difficult to interpret - 16% of the drugs affected the concentration of at least 1 amine. What is the 17%? And what are the black and red blocks below?

We like the overall structure of this figure which is similar to our other publications showing an overview of xenobiotic screens (Fig1A in (Maier et al. 2018); Fig 1C in (Lindell et al. 2024)). However, we take on board the feedback about the lack of clarity and have improved the optical presentation and description of the figure.

Figure 2 - legend B is cryptic, please reformulate

Thank you for the suggestion. We have improved the figure legend:

“(B) Distribution of the number of species against which each compound is effective. While 84% of the compounds do not affect amine output of any species, 8.7% affect two species or more.”

Figure 2C - the legend is back and white but the figure has 3 colors: pink, red and green... please rethink the legend or the figure

We have improved the legend as requested by using a striped pattern instead.

Figure 2D - what is PEA?

PEA is short for 2-phenylethylamine. We have added the definition to the legend.

P13 - Please try to be consistent when reporting units: either mL or ml

Thank you for noticing this. We are now using mL/ μ L throughout.

P13, L25 - avoid reporting rpm as it depends on the size of the centrifuge's rotor

This refers to the settings of a shaker, not a centrifuge, and is therefore the appropriate unit.

P14, L3 - 'cold extraction buffer' implies a buffered solution, that is not the case of what is reported to have been used, as there isn't any buffer capacity in a solution of acetonitrile, methanol, formic acid and two other compounds....

We agree and have changed this to 'cold extraction solution' for technical accuracy.

P14, L9-22 - the details of LC-MS methods: settings, column, MRM transitions per compound, etc need to be reported...

All these details can be found in Supplementary Data 2. Apologies, we had failed to reference this table in the relevant methods section (now added).

P5, L22 - here, one expects reporting of some quantitative values on replicate variability, and what is considered as "high data consistency". What was a minimum and maximum variation, etc? Were some of the replicates excluded due to too high variability - what were the criteria? Given that the authors performed a very large screen and even were able to correct batch effects throughout measurements, as depicted in Fig1S, it would be good to report these "errors".

No metabolomics data points were excluded due to inconsistency. The Methods section, which we have expanded to conform to journal guidelines, accurately describes the data procession. All scripts and raw data are provided in an online repository. Figure EV1 displays all commonly used data quality indicators such as CV, replicate correlation and statistical power calculations.

Reviewer #3:

In this work, the authors implement a high-throughput targeted metabolomics assay to measure amine production of gut bacteria in response to xenobiotics, including pharmaceuticals (antibiotics), pesticides, and other clinically and agriculturally relevant compounds. Leveraging growth data, they compare the relationship between growth rate and amine production in response to different xenobiotics and reveal an interesting, but perhaps not surprising finding, that the production of amines does not necessarily correlate with the magnitude or direction of the impact on growth. These results suggest that the bacteria are executing specific responses to these xenobiotics, and the high degree of responses suggest a greater complexity to this response. The authors go on to show that antibiotics trigger polyamine production in *E. coli*. The review process lends itself to a more critical tone, so I would like to emphasize that the study is valuable and just needs more appropriate statistics and deeper insights to highlight all it has to offer.

What are the key conclusions: specific findings and concepts?

1. Decoupling of growth rate and amine production. In some cases (the proportion varies across species), amine production is significantly altered but not proportional to growth changes. This indicates a more compound-specific stress response is occurring (usually at low doses of the xenobiotic).
2. Amine production follows non-linear response curve to Abx. As xenobiotic dosage increases, there appears to be a burst of amine production at low concentrations that decreases at increased concentrations.
3. Growth responses between related species are more consistent whereas the amine production varies. *E. coli* and *K. aerogenes* show similar growth responses to xenobiotics but *E. coli* overproduces polyamines, whereas *K. aerogenes* does not. This was

What were the methodology and model system used in this study?

The authors implement an impressive high-throughput targeted metabolomics assay to measure amine production (N=6) of four clinically-relevant gut bacteria in response to xenobiotic exposure. They also perform dose-response curves, measuring growth rate and amine production across a range of xenobiotics for some specific cases.

General remarks

- Are you convinced of the key conclusions?

1. Decoupling of growth rate and amine production. The data suggests this is largely true, however, the correlation analyses leading to these conclusion are flawed in their reliance on pearson correlations, which are heavily skewed by outlier data points. Furthermore, a statistical test (e.g. Fishers z-test) between these correlation values can be performed to establish their significance.
2. Amine production follows non-linear response curve to Abx. This is quite clear from figures 3E-G however, may not be surprising. One interpretation of this is that while bacteria can execute these stress responses at low concentrations of xenobiotic, they become overwhelmed at higher concentrations and die out or opt for other response pathways.
3. Growth responses between related species are more consistent whereas the amine production varies. This is evident in figure 3A/B. While I expect these results will hold, they are once again using pearson correlation as the primary metric and lack a definitive statistical test to compare the correlations.

- Place the work in its context

In recent years, there have been several studies on drug-microbe interactions and it is known that these screens lead to the discovery of novel mechanisms with potentially clinical implications. This

is the largest I have seen that combines both growth rate and amine production across multiple clinically-relevant species.

- What is the nature of the advance (conceptual, technical, clinical)?

This advance is primarily technical and really pushes the scale by which we can collect multidimensional data, especially amine production.

- How significant is the advance compared to previous knowledge?

The authors highlight that this is the first pan-xenobiotic scale overview of the relationship. There is a great degree of new, useful, data that could lead to novel hypotheses about drug-microbe-host interactions and new biological mechanisms. While the potential is great, nothing explored or presented does not strike me as a very significant jump. There are many small vignettes and insights drawn in this study but the authors could do better to highlight or emphasize the more impactful findings.

- What audience will be interested in this study?

This study will be of interest to clinicians interested in integrative health solutions that consider the role of the microbiome in drug responses. Microbiologists interested in the gut microbiome will be interested in how amine production in response to low-dose xenobiotics may influence microbial interactions and the overall ecology of the microbiome. There may also be implications for environmental bacteria that may be exposed to low doses of xenobiotics, which would be interesting.

We would like to thank the reviewer for the constructive tone and in-depth analysis of our work. The feedback provided was valuable for improving the manuscript, as outlined below.

Major points

R3.1 - Pearson correlation

Coupling vs decoupling of metabolic responses to growth rate is largely determined using Pearson correlations, which are highly sensitive to outliers. Given that the distribution of responses are biased, with mostly minimal changes and a few extremely strong responses, more robust correlation metrics should be applied (Biweight Mid Correlation (Bicor), Spearman). You may also consider first filtering for only drug-microbe combinations with significant differences in metabolite production before assessing correlations, which would reduce the bias in your distributions. Furthermore, correlations should be reported with p-values. Also, if you are claiming one correlation is stronger than another, a statistical test (Fishers z-score perhaps?) should be performed to support this claim.

The Reviewer raises valid points which we have followed up with additional statistical analysis. As expected, based on the nature of the data and the tests, Pearson and Spearman tests return

different correlation coefficients. Further, the correlation strength differs, as expected, when only the hits are used (see the summary table below). It is important to note that the correlations remain significant in most cases for both Pearson and Spearman. The strength of the correlation is indeed, as pointed out by the reviewer, different between the two tests due to the structure of the data (most compounds having no effect, resulting in a cloud around fold-change 1, and some compounds having strong effects, resulting in a smaller concentration of points around 0). However, it would be misleading to characterize these strongly inhibiting effects as outliers. It is an expected biological observation for a large-scale screen as ours whereby most compounds are not expected to have effect on the readout used.

Species	Responder	HitsOnly			False				True
		Metric	Pearson	Spearman	Kendall	Fraction_ODconcordant	Pearson	Spearman	Kendall
Cs	Tryptamine	0.88***	0.23***	0.16***	0.93	0.87***	0.68***	0.52***	0.86
Ec	Cadaverine	0.22***	-0.07**	-0.05**	0.92	0.53***	0.39***	0.23***	0.34
	Putrescine	0.52***	0.44***	0.32***	0.94	0.43**	0.67***	0.48***	0.44
Ka	Cadaverine	0.42***	0.39***	0.28***	0.95	0.79*	0.52n.s.	0.29n.s.	0.12
	Histamine	0.62***	-0.28***	-0.21***	0.95	0.67***	0.74***	0.55***	0.75
	Putrescine	0.54***	0.06*	0.04*	0.96	0.21n.s.	0.22n.s.	0.17n.s.	0.29
Rg	2-Phenylethylamine	0.7***	0.21***	0.15***	0.86	0.76***	0.78***	0.6***	0.76
	Tryptamine	0.73***	0.22***	0.15***	0.87	0.8***	0.81***	0.62***	0.76

Rebuttal Table 1: Growth-metabolite concentration at the level of individual replicates described by various correlation metrics (relating to Fig 2D-F). Significance keys: ***p<0.001, ** p<0.01, *p<0.05.

With the aim of finding a statistical measure to describe the degree of correlation between metabolite-growth which is both intuitive, accurate and not dependent on the number of hits, we have decided to remove correlation coefficients from **Fig2** (growth-metabolite correlation) and instead describe the fraction of hits which are concordant with growth as a measure of quantifying metabolite-growth coupling. This metric is already introduced earlier (**Fig1C**), making the data presentation more consistent.

In **Fig3A-C** we are investigating growth-growth and metabolite-metabolite correlations between different species. In **FigEV3G-J**, we are investigating the correlation between biological replicates. In both cases, we are working under the implicit null hypothesis that responses are equal in magnitude (and therefore linearly correlated, rather than just a monotonic relationship) and we are interested in the shared occurrences of 'no response', rather than just hits. Pearson correlation is therefore the most appropriate metric. We have added p-values to the coefficients as requested.

R3.2 - Antibiotics rationale

There are many hits it seems yet in figure 4 the authors choose to focus on the least interesting ones with known antibiotic effects that would obviously trigger stress responses. Perhaps the story could be reworked to motivate this instead. Why is it important to know that low-doses of antibiotics activate amine production? I can imagine clinical implications, but these types of rationales for what you investigate deeper should be clearly stated.

We appreciate this comment. It is indeed intuitive that antibiotics trigger polyamine production as a common stress response mechanism. However, this has so far not been described in anaerobic conditions, which is key, since previous studies in aerobic conditions attribute the effect largely to ROS stress. We have added a sentence explaining this knowledge gap and our rationale to the Results section to motivate this part as suggested:

“While it is not unexpected that antibiotics trigger polyamine production (a common stress response mechanism), we were intrigued by the differences between antibiotic classes and the general lack of published data investigating the impact of antibiotics in anaerobic environments.”

R3.3 - Clinical consequences

The impact of this paper hinges on the consequences of the amine production response. That there is a stress response to antibacterials is not very exciting (barring sufficient rationale). Figure 4, focusing on antibacterials would be much more interesting if included a figure like 4C, but with a xenobiotic that is not explicitly antibacterial. Do any of the antidepressants or chemotherapeutics stimulate amine production? How might this amine production influence patient outcomes?

We appreciate this point and have expanded our analysis and discussion of non-antibiotics. We have performed follow up validation experiments with three relevant compounds (new Figure 4D-F): the selective serotonin reuptake inhibitor paroxetine, the HIV drug didanosine and the pesticide fenazaflor. These compounds are relevant due to their widespread and prolonged use, and all three hits were successfully validated. For paroxetine, we were able to link increased cadaverine production to a strong upregulation of *cadA* in the transcriptomics dataset of Ricaurte et al (2024). Furthermore, we now discuss the implication and likely mechanism in detail, particularly with regards to the investigation by Guillen et al (2024) which describes orthogonal mechanisms of non-antibiotic killing.

We now also mention the serum concentration of these amines in the human host and highlight that in particular aromatic amines have potent effects even at low concentrations (Results):

“Robust and substantial production of amines was observed for all amine-bacteria pairs at baseline conditions (DMSO-treated control cultures), with concentrations ranging from 35 to 1100 μ M (Figure EV1F). This production capacity is significant in the context of the nanomolar-range physiological concentrations of these amines in the human body [53–

56] and the ability of aromatic trace amines to exert strong effects at very low concentrations [9,57].”

R3.4 - Dose dependency interpretation

There is a lot of emphasis on the non-monotonous dose dependency of growth and amine production, but it seems like there is a stress response at low doses (as expected by antibiotics), but the cells just die at higher doses. Why is this important or what insights does this give us?

In the context of xenobiotic toxicity on microbiota, this is crucial. Dose-dependency is key in toxicology ('the dose makes the poison') and administering less of a pharmaceutical drug or limiting the intake of pesticides is implicitly assumed to also result in weaker effects on the human-microbiome holobiont. We show here that this might not be true in the context of microbial metabolism. We have now spelled this out more clearly in the Discussion, although we simultaneously acknowledge that this is an *in vitro* study with single bacterial isolates, which may not totally recapitulate microbiota metabolism in the gut.

Another key takeaway message from our study is systematic evidence that xenobiotics have effects beyond the growth of bacteria. While we measure a single class of metabolites, viz. polyamines, in this study, it is likely that the effects might extend to other metabolites as anecdotally observed in some cases before (e.g. for duloxetine in Klunemann et al. 2021). We hope that our study will raise awareness of metabolic effects of xenobiotics and future investigations will consider this in their study design or in mechanistically connecting perturbations to phenotypes. We have added a sentence in Discussion to highlight this.

Our study provides the first pan-xenobiotic scale overview of the relationship between growth and metabolic output in anaerobically cultured gut bacteria, providing systematic evidence that xenobiotics have effects beyond the growth of bacteria. We find that many compounds, particularly antibacterials, can uncouple amine production from growth and cause an increase in amine production in *E. coli*. Polyamines are well-known stress response metabolites in the

R3.5 - Metabolite growth correlation

The argument that metabolite production is coupled to growth seems to overlook the fact that fewer cells will produce less metabolites. I did not see a description in the methods of how measured production would be normalized to absolute growth rate or biomass. As is, it suggests that in these cases the cells adjust production in response to xenobiotic exposure in an amount that is commensurate to the effect on growth rate. Another interpretation is that there are just fewer cells so production is lower.

Metabolite-growth coupling is in a way a null expectation as the reviewer suggests; hence, we highlight cases where this does not hold. Further, this correlation would be expected only in the

case of metabolite productions that are coupled to growth due to energetics/mass balance, which is not the case for amines.

We did not normalize metabolite concentrations to cell number/mass and do not investigate the rate of growth or metabolite production. We focus on endpoint measurements of growth and production which enabled us to perform this study at an unprecedented scale/number of compounds. A compound is classified as a hit if amine abundance is lower or higher compared to DMSO controls. In many cases (the growth concordant cases), this will be due to impaired growth, resulting in fewer cells in the culture and therefore fewer metabolites. Normalization of production levels by growth will not yield additional insights since this is considered in our classification of growth dependent/independent change in production.

We have added a sentence to the Results section to clarify this:

“Growth inhibition results in lower bacterial biomass with an expected concurrent reduction in metabolic activity. “

-Specify experiments or analyses required to demonstrate the conclusions

1) Recalculate correlations with more robust correlation metrics and appropriate statistical tests

This has been done, as described above.

2) I would really like to see 1) More examples of surprising or new insights or 2) mechanisms of bacterial responses in this work. As is, it's clear a lot of data was generated but I don't think any especially surprising or interesting results were reported. I am sure they are there. Either of these two analyses would cement the value of the data generated in this work and how it influences clinical work or improves our understanding of microbial responses to xenobiotics.

For surprising results:

1) More investigations into amine production in response to non-antibacterials. The authors could highlight drugs that are not explicitly antibacterials, but still elicit amine production. Additional discourse on how these amines might modulate host physiology or clinical outcomes would be a huge plus. You have already identified many xenobiotics that stimulate metabolite production, but don't affect growth (2F), so this should be relatively straightforward.

This has been done for three relevant and diverse compounds, namely paroxetine, didanosine and fenazaflor. For paroxetine, we provide additional mechanistic insights derived from re-analyzing a published dataset by Ricaurte et al (new Figure 4G-H).

For mechanisms:

2) Are there different pathways regulating production of these amines? What regulatory programs are controlling these responses and are they shared between different classes of xenobiotics? A few examples of RNA-Seq in response to xenobiotic exposure or or RT-qPCR comparing these

responses between different classes of xenobiotics would be nice to see. Perhaps other studies such as (<https://doi.org/10.1038/s41564-023-01581-x>) can provide this data.

We have mined the dataset by Ricaurte et al and found strong upregulation of cadaverine biosynthesis and the acid stress response gene *asr* under SSRI treatment (new Figure 4G-H). We have expanded the Discussion around the mechanisms of antibiotic and non-antibiotic killing.

-Motivate your critique with relevant citations and argumentation

As highlighted in your introduction, stress responses triggers polyamine production in several species. The paper mentioned above (<https://doi.org/10.1038/s41564-023-01581-x>) is a nice demonstration of high-throughput screens with follow up studies to establish new mechanistic insights. Granted, transcriptional data is more suited for this, but RNA-seq in *E. coli* is a fairly straightforward pipeline that could be applied here.

We have analyzed the Ricaurte et al. dataset. This has further strengthened our conclusions with support from an independent study.

R3 - Minor points

-Easily addressable points

1) It would be nice to have a table describing the amines you measure, and perhaps a brief description of their clinical relevance.

Please see Fig 1A.

2) Correlations should include p-values

We have improved the use of correlation metrics throughout the manuscript (see above) and report p-values.

3) Page 10, line 2. Figure 4 is about cadaverine, it would be nice to have a rationale for why this is important. With all the data, why focus on this story?

Thank you for the feedback, we have added the following sentence:

“E. coli showed by far the most instances of increased metabolite production and the vast majority of these (67/85) were cadaverine. “

4) Page 10, Line 16: You highlight bacteriostatic antibiotics despite not having data for any. Unless you have data for bacteriostatic abx, I would suggest removing this as it sets up expectations. Line 18 describes how you found no link since all B-lactams were cidal, which is a poor statement to make since you didn't test any bacteriostatic compounds. Bacteriostatic vs -cidal and amine production would have been really interesting though.

Our analyses (Fig EV3) include tetracycline, a classic bacteriostatic antibiotic inhibiting protein synthesis. We show that tetracycline-treated cultures (unlike for beta-lactams) contain live CFUs but show no spike in polyamine production. Of course this is just one compound, but we do not feel that a more comprehensive analysis would add to the main conclusions of the manuscript.

5) Page 13, line 41: What did you do with the data in cases where the OD was confounded by the aggregation?

Apologies for leaving out this important detail. OD values with a fold-change of >1.5 were excluded from the analysis. We have added this information to the Methods. As stated, this affects a small number of samples (353 out of 4608, 7.7%) and it does not affect the metabolomic analyses.

6) Page 11 line 23-25: I don't really understand the latter part of this sentence.

The Reviewer is referring to: “and show that polyamines are stress-responsive in the context of anaerobic metabolism.” This point relates to the fact that the literature has so far attributed polyamine induction by antibiotic stress to reactive oxygen species, which are not expected to occur in anaerobic conditions. Another molecular mechanism must therefore link antibiotic exposure to polyamine production in anaerobic conditions. We have clarified the sentence (“in the context of anaerobic metabolism (in the absence of reactive oxygen species)”) and now discuss the possible mechanisms in more detail.

“What is the mechanism causing polyamine upregulation in response to antibiotic exposure under anaerobic conditions? There remain significant knowledge gaps around the downstream mechanisms of action of bactericidal antibiotics in anaerobic conditions, specifically how inhibiting the direct molecular target (penicillin-binding proteins in the case

of β -lactams) eventually leads to cell death. Reactive electrophilic species have been described to play a role in this context, with downstream effects including DNA and membrane damage [76], which in turn could induce polyamines.”

-Presentation and style

1) 2C: Instead of dark and light colors, striped and solid could be more clear.

Good suggestion, we have adapted the figure accordingly.

2) 2G is a bit hard to read. You could add a gap in between the columns between different amines.

Good suggestion, we have added guiding lines to the heatmap.

-Trivial mistakes

1) 3B and 3C are not mentioned in the text

Thank you for spotting this. We have adapted the text.

2) 3G is not mentioned in the text or the legend, despite being an important figure and quite possibly one of the highlights of this work.

Thank you for spotting this. We have adapted the text and legend.

3) 3E/F Should indicate that this is *R. gnavus* in the legend or figure.

These panels show data for *E. coli*, which had indeed not been described/labeled properly. We have added this information.

4) Page 12, line 6: Italicize and correct spelling for Enterobacteriaceae

This has been corrected as suggested.

5) Page 14, line 24: Python version is incorrect here

Thank you, this has been corrected.

2nd Jun 2025

Manuscript Number: MSB-2024-12833R

Title: Impact of drugs and environmental contaminants on amine production by gut bacteria

Dear Prof Patil,

Thank you for the submission of your revised manuscript to Molecular Systems Biology. We have now received the enclosed reports from the referees that were asked to re-assess it. As you will see the reviewers are now globally supportive and I am pleased to inform you that we will be able to accept your manuscript pending the following final amendments:

- 1) Please submit the main manuscript file as a .docx file with no track changes and the figures removed. Alternatively you may submit the manuscript in LaTeX format.
- 2) please include keywords to max. 5.
- 3) Please merge the Code Availability into the Data availability section and format according to the example below:
"The datasets and computer code produced in this study are available in the following databases:
- Chip-Seq data: Gene Expression Omnibus GSE46748 (<https://www.ncbi.nlm.nih.gov/geo/query/acc.cgi?acc=GSE46748>)
- Modeling computer scripts: GitHub (<https://github.com/SysBioChalmers/GECKO/releases/tag/v1.0>)
- [data type]: [full name of the resource] [accession number/identifier] ([doi or URL or identifiers.org/DATABASE:ACCESSION])"
- 4) Please include a README file on Mendeley Data with practical use instructions for potential future users of your code and ensure that the Raw Data and Code is fully published with an active DOI.
- 5) Please include a "Disclosure and competing interests statement". We updated our journal's competing interests policy in January 2022 and request authors to consider both actual and perceived competing interests. Please review the policy <https://www.embopress.org/competing-interests> and update your competing interests if necessary.
- 6) References: Please correct the reference citation in the reference list to be alphabetical (not numerical). Where there are more than 10 authors on a paper, only the first 10 should be listed, followed by "et al.". Please check "Author Guidelines" for more information.
<https://www.embopress.org/page/journal/17574684/authorguide#referencesformat>
- 7) Our journal encourages inclusion of *data citations in the reference list* to directly cite datasets that were re-used and obtained from public databases. Data citations in the article text are distinct from normal bibliographical citations and should directly link to the database records from which the data can be accessed. In the main text, data citations are formatted as follows: "Data ref: Smith et al, 2001" or "Data ref: NCBI Sequence Read Archive PRJNA342805, 2017". In the Reference list, data citations must be labeled with "[DATASET]". A data reference must provide the database name, accession number/identifiers and a resolvable link to the landing page from which the data can be accessed at the end of the reference. Further instructions are available at .
- 8) In the Methods, please take care of the following:
 - The Materials and Methods section should be renamed to "Methods".
 - Please ensure that a statement on whether or not blinding was done is included in the Methods even if no blinding was done. Please also be sure to update the Author Checklist with this information and where it can be found in the manuscript.When submitting your revised manuscript, please do not include the Reagents and Tools Table in the Methods section of the manuscript but upload it as a separate file choosing the file type "Reagent Table". Please also ensure that the file is made using our template, which you can find in our author guidelines:
<https://www.embopress.org/page/journal/14693178/authorguide#structuredmethods>.
- 9)
- 10) Please place individual sections of the manuscript in the following order: Title page - Abstract & Keywords - Introduction - Results - Discussion - Methods - Data Availability - Acknowledgements - Disclosure and Competing Interests Statement - References - Figure Legends - Expanded View Figure Legends.
- 11) For the figures and figure legends, please take care of the following:
 - Please remove all figures from main manuscript file and leave only main figure legends placed after the references.
 - Please note that the legend for figure 4D-H is missing in the manuscript. This needs to be rectified.
 - Please note that the exact p values are not provided in the legends of figures 3A-C; EV1 H,
 - Please indicate the statistical test used for data analysis in the legends of figures 1C, 3A-D; EV2, EV3 C
 - Please note that scale bar and its definition are missing for figures EV4 B, C
- 12) Dataset EV: The source file names and titles of the Dataset EV files all need to be updated to Dataset EV1-EV4 instead of Supplementary Data. The files should be uploaded individually as Dataset files with legends in a separate tab/sheet in each Excel file
- 13) Appendix file: Please upload the Appendix as a single PDF (no separate image files are needed).
- 14) Synopsis:
 - Synopsis image: Please remove it from the manuscript and upload it as a high-resolution jpeg or png file 550 pixels wide x (300-600) pixels high. Currently the format is eps and the image is smaller than our dimensions.
 - Synopsis text: Please remove from the main manuscript file and only supply the separate .doc file. In addition to the standfirst

text, please include max 5 bullet points. Please write the bullet points to summarise the key NEW findings. They should be designed to be complementary to the abstract - i.e. not repeat the same text. We encourage inclusion of key acronyms and quantitative information (maximum of 30 words / bullet point). Please use the passive voice.

15) As part of the EMBO Publications transparent editorial process initiative (see our policy here:

https://www.embopress.org/transparent-process#Review_Process), Molecular Systems Biology will publish online a Peer Review File (PRF) to accompany accepted manuscripts. This file will be published in conjunction with your paper and will include the anonymous referee reports, your point-by-point response and all pertinent correspondence relating to the manuscript. Let us know whether you agree with the publication of the PRF and as here, if you want to remove or not any figures from it prior to publication. Please note that the Authors checklist will be published at the end of the PRF.

16) After your paper is published, we may promote it on social media. If you have any handles or hashtags for Bluesky you would like included, please let us know.

17) Please provide a point-by-point letter INCLUDING my comments as well as the reviewer's reports and your detailed responses (as Word file).

I look forward to reading a new revised version of your manuscript as soon as possible.

Yours sincerely,

Poonam Bheda, PhD
Scientific Editor
Molecular Systems Biology

Reviewer #1:

The authors have done a great job in addressing the comments and concerns from my review. I am fully satisfied with their responses to my queries, as well as other reviewer queries. The revised manuscript is very clear to read, with the data presented very well.

Reviewer #2:

The authors have made an important effort in taking the comments of the reviewers into consideration, which greatly improved this revised version of the manuscript. Therefore I do not have further points to comment. My recommendation to the editors is that this manuscript is worthy of publication in MSB.

Reviewer #3:

I thank the authors for providing comprehensive revisions and addressing all my concerns. In particular, the additional analysis incorporating transcriptional and knockout data from previous studies provides a nice mechanistic link connecting xenobiotics to polyamine biosynthesis. I also appreciate the substantial changes to reported statistics. The removal of correlation metrics for some plots is fair given the intent of the figure to show 'rare' events which are not necessarily outliers. Please see my few remaining suggestions.

Figure 2E legend - It still says "The pearson r is shown above the plot". Remove this.

Page 8, lines 18-21: P values should be reported here, not just *** in the figure

Figure 4D - 4H don't have legends?

Page 11, line 11-12: "This is strong evidence..." I think this summary statement is missing the main point, which is in my opinion to support the data that paroxetine exposure is driving cadverine production as well as provide a mechanistic link.

Page 10 line 34-35: What is the significance of speA/B genes here & relevance to polyamine biosynthesis?

Figure EV4A) should have pearson r and p-value

We would like to thank all reviewers and the editor for the detailed, constructive feedback. We have addressed final suggestions and editorial requirements as outlined below.

Beyond the requested changes outlined below, have changed 'non-linear' to 'non-monotonic' to describe the dose-dependency as this more accurately describes the mathematical relationship.

Additionally, we now additionally cite a relevant recent paper by Chen et al (PMID: 40348778) in the Introduction which has been published since the last submission.

Previous large-scale studies on xenobiotic-bacteria interactions have focused on growth effects (Maier *et al*, 2018; Lindell *et al*, 2024). Only a few studies have systematically investigated effects on bacterial physiology, notably one study investigating transcriptomic responses to >400 drugs (Ricaurte *et al*, 2024) ~~and another investigating metabolic responses to 18 pesticides~~ (Chen *et al*, 2025). ~~Yet, neither of these approaches capture the impact on production levels of health-linked metabolites since the pathway from the transcriptome to metabolome is complex and non-linear. Therefore, w~~We here investigated the effect of a large number of xenobiotics on gut bacterial amine metabolic output. Our results provide the first large-scale, pan-xenobiotic, map of xenobiotic-species-metabolite interactions and highlight the potential of diverse xenobiotics to interfere with amine production by gut bacteria.

1) Please submit the main manuscript file as a .docx file with no track changes and the figures removed. Alternatively you may submit the manuscript in LaTeX format.

Done.

2) please include keywords to max. 5.

Done.

Microbiome, xenobiotics, pesticides, stress response, polyamines

3) Please merge the Code Availability into the Data availability section and format according to the example below:

"The datasets and computer code produced in this study are available in the following databases:

- Chip-Seq data: Gene Expression Omnibus GSE46748
(<https://www.ncbi.nlm.nih.gov/geo/query/acc.cgi?acc=GSE46748>)

- Modeling computer scripts: GitHub
(<https://github.com/SysBioChalmers/GECKO/releases/tag/v1.0>)

- [data type]: [full name of the resource] [accession number/identifier] ([doi or URL or identifiers.org/DATABASE:ACCESSION])"

The sections have been merged, and the cited repository has been made public.

4) Please include a README file on Mendeley Data with practical use instructions for potential future users of your code and ensure that the Raw Data and Code is fully published with an active DOI.

We have added readme-style information to the repository description. The repository is now public with DOI: 10.17632/cds7tvdb85.1. This information has been updated in the manuscript.

5) Please include a "Disclosure and competing interests statement". We updated our journal's competing interests policy in January 2022 and request authors to consider both actual and perceived competing interests. Please review the policy <https://www.embopress.org/competing-interests> and update your competing interests if necessary.

We have reviewed the policy and declare no competing interests. The statement is now included in the manuscript.

6) References: Please correct the reference citation in the reference list to be alphabetical (not numerical). Where there are more than 10 authors on a paper, only the first 10 should be listed, followed by "et al.". Please check "Author Guidelines" for more information.

<https://www.embopress.org/page/journal/17574684/authorguide#referencesformat>

We have changed the referencing format as requested.

7) Our journal encourages inclusion of *data citations in the reference list* to directly cite datasets that were re-used and obtained from public databases. Data citations in the article text are distinct from normal bibliographical citations and should directly link to the database records from which the data can be accessed. In the main text, data citations are formatted as follows: "Data ref: Smith et al, 2001" or "Data ref: NCBI Sequence Read Archive PRJNA342805, 2017". In the Reference list, data citations must be labeled with "[DATASET]". A data reference must provide the database name, accession number/identifiers and a resolvable link to the landing page from which the data can be accessed at the end of the reference. Further instructions are available at

[Th<https://www.embopress.org/page/journal/17574684/authorguide#referencesformat>](https://www.embopress.org/page/journal/17574684/authorguide#referencesformat).

We have reviewed the guidelines. We do re-use previously published datasets from two primary sources publications (Ricaurte et al, and Noto Guillen et al). However, since the data was obtained from supplementary tables in these publications and not from public databases, our interpretation of the guidelines is that these should be cited in the usual way.

8) In the Methods, please take care of the following:

- The Materials and Methods section should be renamed to "Methods".

Done.

- Please ensure that a statement on whether or not blinding was done is included in the Methods even if no blinding was done. Please also be sure to update the Author Checklist with this information and where it can be found in the manuscript.

We have added a statement to Methods, section "Growth data analysis": "No blinding was done in the study". The Author Checklist has been updated.

When submitting your revised manuscript, please do not include the Reagents and Tools Table in the Methods section of the manuscript but upload it as a separate file choosing the file type "Reagent Table". Please also ensure that the file is made using our template, which you can find in our author guidelines:

Reagents and Tools Table is now a separate file and adheres to template.

9)

Point 9 missing?

10) Please place individual sections of the manuscript in the following order: Title page - Abstract & Keywords - Introduction - Results - Discussion - Methods - Data Availability - Acknowledgements - Disclosure and Competing Interests Statement - References - Figure Legends - Expanded View Figure Legends.

Done.

11) For the figures and figure legends, please take care of the following:

- Please remove all figures from main manuscript file and leave only main figure legends placed after the references.

Done.

- Please note that the legend for figure 4D-H is missing in the manuscript. This needs to be rectified.

Apologies for this omission, now rectified.

- (D) – (F) Dose-response curves for fenazaflo^r (pesticide), didanosine (antiviral drug) and paroxetine (antidepressant) indicating fold-changes of either metabolite concentrations (red and purple) or growth at stationary phase, compared to untreated controls. Lines indicate the mean and shaded areas indicate the standard deviation of n=8 biological replicates.
- (G) Re-analysis of data from (Ricaurte *et al*, 2024) illustrating transcriptomic regulation of polyamine biosynthesis genes in *E. coli* MG1655 in the presence of different selective serotonin reuptake inhibitor (SSRI) antidepressant drugs. Data was obtained from Supplementary Table 8.
- (H) Volcano plot illustrating genome-wide transcriptional changes in *E. coli* MG1655 in the presence paroxetine, obtained from Supplementary Table 8 of (Ricaurte *et al*, 2024). Differentially abundant transcripts with known roles in polyamine metabolism are annotated in the plot. Red lines indicate the fold-change thresholds used in the original publication ($\log_2(\text{fold-change}) > 1$ and $p_{\text{adj}} < 0.05$ (FDR-corrected p-values obtained via DESeq2)).

- Please note that the exact p values are not provided in the legends of figures 3A-C; EV1 H,

We have added p-values in Fig3 as requested. In Fig EV1 H (a theoretical calculation of statistical power), the lines labelled 'p<0.05' etc indicate the p-value threshold relevant to the study, therefore this labelling is appropriate.

- Please indicate the statistical test used for data analysis in the legends of figures 1C, 3A-D; EV2, EV3 C

Apologies, we have added this information.

- Please note that scale bar and its definition are missing for figures EV4 B, C

We have added scale bars and their definition.

12) Dataset EV: The source file names and titles of the Dataset EV files all need to be updated to Dataset EV1-EV4 instead of Supplementary Data. The files should be uploaded individually as Dataset files with legends in a separate tab/sheet in each Excel file

We now provided these tables as individual files with updated names as requested.

13) Appendix file: Please upload the Appendix as a single PDF (no separate image files are needed).

There is no appendix associated with this manuscript.

14) Synopsis:

- Synopsis image: Please remove it from the manuscript and upload it as a high-resolution jpeg or png file 550 pixels wide x (300-600) pixels high. Currently the format is eps and the image is smaller than our dimensions.

We have changed and expanded the synopsis image to better reflect the experimental approach taken in the paper. It is now provided as png in the correct dimensions (550x553).

The lower right corner would be a suitable thumbnail for the website

- Synopsis text: Please remove from the main manuscript file and only supply the separate .doc file. In addition to the standfirst text, please include max 5 bullet points. Please write the bullet points to summarise the key NEW findings. They should be designed to be complementary to the abstract - i.e. not repeat the same text. We encourage inclusion of key acronyms and quantitative information (maximum of 30 words / bullet point). Please use the passive voice.

We now provide the synopsis in a separate file only and have expanded it with bullet points.

The impact of >1,700 xenobiotics on growth and amine metabolism of 4 gut bacteria was screened *in vitro*. Xenobiotics, in particular antibiotics, disrupt metabolic homeostasis in both growth dependent and independent manner, often displaying non-linear dose-dependency.

- 275 xenobiotic compounds affect amine metabolism across 747 xenobiotic-species-metabolite interactions
- In one third of these interactions, metabolic changes are decoupled from growth
- *Escherichia coli* produces polyamines in response to diverse antibiotic and non-antibiotic compounds
- Metabolic responses display unusual dose responses with weaker doses often having stronger effects

We have checked the text and images.

15) As part of the EMBO Publications transparent editorial process initiative (see our policy here: https://www.embopress.org/transparent-process#Review_Process), Molecular Systems Biology will publish online a Peer Review File (PRF) to accompany accepted manuscripts. This file will be published in conjunction with your paper and will include the anonymous referee reports, your point-by-point response and all pertinent correspondence relating to the manuscript. Let us know whether you agree with the publication of the PRF and as here, if you want to remove or not any figures from it prior to publication. Please note that the Authors checklist will be published at the end of the PRF.

We agree with the publication of the peer review in full.

16) After your paper is published, we may promote it on social media. If you have any handles or hashtags for Bluesky you would like included, please let us know.

@skamrad.bsky.social

@kiranrpatil.bsky.social

17) Please provide a point-by-point letter INCLUDING my comments as well as the reviewer's reports and your detailed responses (as Word file).

Done.

Reviewer #1:

The authors have done a great job in addressing the comments and concerns from my review. I am fully satisfied with their responses to my queries, as well as other reviewer queries. The revised manuscript is very clear to read, with the data presented very well.

Thank you.

Reviewer #2:

The authors have made an important effort in taking the comments of the reviewers into consideration, which greatly improved this revised version of the manuscript. Therefore I do not

have further points to comment. My recommendation to the editors is that this manuscript is worthy of publication in MSB.

Thank you.

Reviewer #3:

I thank the authors for providing comprehensive revisions and addressing all my concerns. In particular, the additional analysis incorporating transcriptional and knockout data from previous studies provides a nice mechanistic link connecting xenobiotics to polyamine biosynthesis. I also appreciate the substantial changes to reported statistics. The removal of correlation metrics for some plots is fair given the intent of the figure to show 'rare' events which are not necessarily outliers. Please see my few remaining suggestions.

Thank you for your efforts and constructive, detailed feedback throughout the process. We have addressed these remaining points as suggested.

Figure 2E legend - It still says "The pearson r is shown above the plot". Remove this.

Thank you for noticing this, we have corrected the error.

Page 8, lines 18-21: P values should be reported here, not just *** in the figure

We have added exact p-values to the figures.

Figure 4D - 4H don't have legends?

Apologies for this omission, now rectified.

- (D) – (F) Dose-response curves for fenazaflo^r (pesticide), didanosine (antiviral drug) and paroxetine (antidepressant) indicating fold-changes of either metabolite concentrations (red and purple) or growth at stationary phase, compared to untreated controls. Lines indicate the mean and shaded areas indicate the standard deviation of n=8 biological replicates.
- (G) Re-analysis of data from (Ricaurte *et al*, 2024) illustrating transcriptomic regulation of polyamine biosynthesis genes in *E. coli* MG1655 in the presence of different selective serotonin reuptake inhibitor (SSRI) antidepressant drugs. Data was obtained from Supplementary Table 8.
- (H) Volcano plot illustrating genome-wide transcriptional changes in *E. coli* MG1655 in the presence paroxetine, obtained from Supplementary Table 8 of (Ricaurte *et al*, 2024). Differentially abundant transcripts with known roles in polyamine metabolism are annotated in the plot. Red lines indicate the fold-change thresholds used in the original publication ($\log_2(\text{fold-change}) > 1$ and $p_{\text{adj}} < 0.05$ (FDR-corrected p-values obtained via DESeq2)).

Page 11, line 11-12: "This is strong evidence..." I think this summary statement is missing the main point, which is in my opinion to support the data that paroxetine exposure is driving cadverine production as well as provide a mechanistic link.

We agree and have changed the sentence to the following:

"This is in alignment with the observed metabolic phenotype and provides a mechanistic link via a transcriptional stress response."

Page 10 line 34-35: What is the significance of speA/B genes here & relevance to polyamine biosynthesis?

Thank you, we have added this information:

“Surprisingly, mutants of *speA* (arginine decarboxylase) and *speB* (agmatinase), enzymes producing agmatine and putrescine, often have significantly improved fitness”

Figure EV4A) should have pearson r and p-value

Thank you, we have added this information.

16th Jun 2025

Manuscript number: MSB-2024-12833RR

Title: Impact of drugs and environmental contaminants on amine production by gut bacteria

Dear Prof Patil,

Thank you again for sending us your revised manuscript. We are now satisfied with the modifications made and I am pleased to inform you that your paper has been accepted for publication.

Yours sincerely,

Sincerely,

Poonam Bheda, PhD
Scientific Editor
Molecular Systems Biology
